# Dopamine regulates decision thresholds in human reinforcement learning in males

Karima Chakroun[1], Antonius Wiehler [2], Ben Wagner [3], David Mathar [4], Florian Ganzer [5], Thilo van Eimeren[6], Tobias Sommer[1] & Jan Peters [1,4] ✉

Dopamine fundamentally contributes to reinforcement learning, but recent accounts also suggest a contribution to specific action selection mechanisms and the regulation of response vigour. Here, we examine dopaminergic mechanisms underlying human reinforcement learning and action selection via a combined pharmacological neuroimaging approach in male human volunteers ($n = 31$, within-subjects; Placebo, 150 mg of the dopamine precursor L-dopa, 2 mg of the D2 receptor antagonist Haloperidol). We found little credible evidence for previously reported beneficial effects of L-dopa vs. Haloperidol on learning from gains and altered neural prediction error signals, which may be partly due to differences experimental design and/or drug dosages. Reinforcement learning drift diffusion models account for learning-related changes in accuracy and response times, and reveal consistent decision threshold reductions under both drugs, in line with the idea that lower dosages of D2 receptor antagonists increase striatal DA release via an autoreceptor-mediated feedback mechanism. These results are in line with the idea that dopamine regulates decision thresholds during reinforcement learning, and may help to bridge action selection and response vigor accounts of dopamine.

The neurotransmitter dopamine (DA) plays a central role in a range of cognitive and motivational processes, including cognitive control[1], reinforcement learning[2] and decision-making[3]. Phasic responses of midbrain dopamine neurons encode reward prediction errors, the discrepancy between obtained and expected reward[2]. Prediction error signals play a central role in formal reinforcement learning theory[4,5]. On a neural level, positive prediction errors are thought to be signaled by phasic burst firing of DA neurons[2], predominantly activating low-affinity striatal D1 receptors in the direct pathway that facilitates *go* learning[6–8]. In contrast, negative prediction errors are thought to be signaled by phasic dips of DA neuron firing rates below baseline[2], predominantly affecting high affinity striatal D2 receptors in the indirect pathway that facilitates *no go* learning[6–8].

Target regions of midbrain dopaminergic projections[9] in dorsal and ventral striatum reliably exhibit activation patterns in functional neuroimaging studies that correspond to reward prediction error coding[10,11]. Likewise, animal work has shown a causal role of dopamine neuron signaling in reinforcement learning[12]. On the other hand, causal evidence for a role of DA in human reinforcement learning is primarily based on pharmacological work and studies in patients with dysfunctions in the dopamine system, e.g. Parkinson's disease (PD).

More generally, the contributions of dopamine to reinforcement learning have focused on two aspects, learning (value updating) and performance (action selection)[13,14]. With respect to learning, behavioral and neural effects of pharmacological manipulation of DA appear heterogeneous, replications of specific effects are scarce[15] and many studies suffer from small sample sizes[15]. Elevation of dopamine

[1]Institute for Systems Neuroscience, University Medical Center Hamburg-Eppendorf, Hamburg, Germany. [2]Motivation, Brain and Behavior Lab, Paris Brain Institute (ICM), Pitié-Salpêtrière Hospital, Paris, France. [3]Chair of Cognitive Computational Neuroscience, Technical University Dresden, Dresden, Germany. [4]Department of Psychology, Biological Psychology, University of Cologne, Cologne, Germany. [5]Integrated Psychiatry Winterthur, Winterthur, Switzerland. [6]Multimodal Neuroimaging Group, Department of Nuclear Medicine, University Medical Center Cologne, Cologne, Germany. ✉e-mail: jan.peters@uni-koeln.de

transmission via the DA precursor L-dopa has been suggested to improve in particular *go* learning in healthy participants[16,17] and PD patients[7,18]. Such effects might be driven by enhanced neural reward prediction error responses under L-dopa[15–17]. However, other studies did not find increased striatal reward prediction error responses following L-dopa administration[19,20], and one study even observed increased punishment-related striatal responses[21]. Other evidence suggests blunted prediction error responses following L-dopa administration in PD patients[22]. Conversely, D2 receptor antagonists have sometimes been reported to impair reinforcement learning[17,23], whereas other studies have reported no effects[24], or effects restricted to post-learning decision-making[25,26]. Interpretation of D2 receptor antagonist effects are complicated by the fact that lower dosages predominantly affect presynaptic D2 autoreceptors[27], which, via an inhibition of negative feedback[27], likely increases (rather than decreases) striatal DA release[28–32]. In line with this idea, lower dosages of D2 receptor antagonists in some cases improve learning from positive feedback[8] (similar to some reported effects of L-dopa[17]), enhance prediction error signaling[25] and increase overall stimulus-locked striatal responses[33]. Higher dosages might lead to the reverse effects[23]. This is consistent with attenuated *go* learning and reduced striatal prediction error responses in schizophrenia patients who receive antipsychotics[13].

But dopamine may also critically contribute to action selection per se. One account suggests that increased striatal DA availability during choice increases activation in the striatal *go* pathway, and reduces activation in the *no go* pathway[13,14], thereby facilitating action initiation vs. inhibition. This resonates with accounts that emphasize a role of DA in regulating response vigor[34–37]. This account is conceptually related to a recent proposal that striatal DA regulates decision thresholds during action selection[38], which is also supported by basal ganglia circuit models[39]. It is also related to theoretical accounts emphasizing a role for DA in encoding the (subjective) precision of actions and/or policies[40–42]. Decision thresholds play a central role in sequential sampling models[43] such as the drift diffusion model[44]. In these models, choice behavior arises from a noisy evidence accumulation process that terminates as soon as the accumulated evidence exceeds a threshold. Decision threshold adjustments are thought to be at least in part regulated by basal ganglia circuits[39,45–51]. In line with a more specific role of DA[38,39], rodent response time distributions following amphetamine infusion in the striatum change in a manner that is consistent with a threshold reduction[52]. Likewise, increased DA availability in mice increases response rates in the absence of learning[53], again consistent with a threshold reduction. In humans, administration of the catecholamine precursor tyrosine reduces decision thresholds across different value-based decision-making tasks[54], and the DA agonist ropinirole reduces decision thresholds during inhibition[55]. In contrast, Bromocriptine (a DA agonist) does not affect thresholds during perceptual decision-making[56], suggesting that such effects might be task-dependent. Finally, gambling disorder, a putatively hyperdopaminergic disorder[57], is associated with altered adjustment of decision thresholds over the course of learning[58]. However, direct causal evidence for the role of DA in regulating decision-thresholds in human reinforcement learning is still lacking.

The present study is part of a larger project from which a different learning task has been published previously[19]. The study was initially conceptualized as a replication of the gain condition of Pessiglione et al.[17]. In that study, participants receiving L-dopa (n = 13) showed improved learning from rewards, and enhanced striatal coding of positive vs. negative prediction errors, compared to participants receiving Haloperidol (n = 13). We scanned a larger sample (n = 31) using fMRI, and employed a within-subjects design using slightly higher drug dosages (L-dopa: 150 mg, Haloperidol: 2 mg). Because the primary behavioral effect in the original study was observed in the gain condition, the loss and neutral conditions from the original study were omitted here. Note that in particular the isolation of the gain condition likely contributed to our unsuccessful replication of behavioral and neural effects (see discussion). Although replication was the initial goal of this project, recent developments of combined reinforcement learning drift-diffusion models[59–63] (RLDDMs) allowed us to leverage this data set to examine dopamine effects on action selection. Thus, we jointly tested DA accounts of learning (replication analysis) and action selection (de-novo analysis) within the same data set. First, we aimed to test (and replicate) previously reported effects of DA on reinforcement learning and associated neural prediction error responses[17]. Second, we directly tested the potential role of DA in regulating decision thresholds during reinforcement learning[38,52,54] by leveraging hierarchical Bayesian RLDDMs, and by directly testing for drug effects on model parameters in a combined model across all three drug conditions.

Here we show that pharmacological increases in dopamine neurotransmission result in reductions in decision thresholds (reduced boundary separation parameter) in a stationary reinforcement learning task. This effect was observed across a range of RLDDM modeling schemes, and resonates with recent accounts of the contribution of DA to action selection[38] and the regulation of response vigour[36,37,42].

## Results

Healthy male volunteers (n = 31) performed a stationary reinforcement learning task under three drug conditions (Placebo, L-dopa (150 mg) and Haloperidol (2 mg), double-blind, counterbalanced) during functional magnetic resonance imaging (fMRI), following completion of a previously published reinforcement learning task[19].

### Model-agnostic analysis

Participants failed to respond within the allocated response time window on average less than once (mean [range] number of misses Placebo: .61 [0 – 10], L-dopa: .61 [0 – 7], Haloperidol: .45 [0 – 4]). RT distributions per drug condition are shown in Fig. 1a–c, with choices of the suboptimal option coded as negative RTs. A Bayesian signed rank test[64] provided substantial evidence for performance above chance level under all drugs (see Fig. 1d, all $BF_{10} > 4000$). Summary descriptive statistics for model-agnostic performance measures (accuracy, total rewards earned, median RTs) are shown in Fig. 1d–f and Table 1, and results from Bayesian repeated measures ANOVAs[64] are listed in Table 1. Numerically, both accuracy and median RTs were higher under Placebo compared to both L-dopa and Haloperidol, but Bayesian repeated measures ANOVAs with a within-participant factor of drug and covariates of linear and quadratic effects of working memory capacity as well as linear effects of body weight revealed inclusion Bayes Factors ($BF_{incl}$) < 3 for all effects (Table 1).

Given the within-subjects design, we also tested for a potential meta-learning effect across sessions. This analysis revealed moderate evidence in favour of the null model without a session effect (Supplemental Figure S1, BF01 = 8.41).

To directly replicate the analysis of the primary behavioral effect reported in Pessiglione et al.[17] (difference in total rewards earned between L-dopa and Haloperidol conditions), a two-tailed frequentist Wilcoxon signed-rank test between L-dopa and Haloperidol was conducted (as the assumption of normality was not met, according to Shapiro-Wilk test), which revealed little credible evidence of a difference between conditions (z = .668, p = .510, r = .140, 95% CI = [−.263, .501]). This was also the case when restricting the analysis to only those participants who received L-dopa and Haloperidol on the first session via a two-tailed frequentist two-sample t-test (assumption of normality was met, according to Shapiro-Wilk test, $t_{19}$ = −.943, p = .358, Cohen's d = .412, 95%CI = [−.459, 1.273]), which shows that, even in participants where meta-learning across sessions can be ruled out, little credible evidence for a performance difference between L-dopa and Haldol was found.

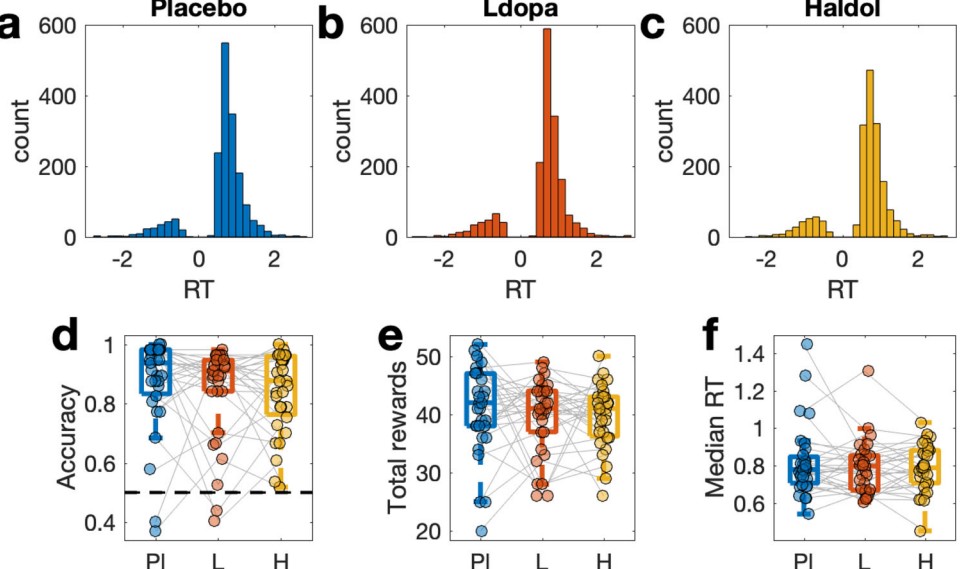

**Fig. 1 | Behavioral data.** Response time (RT) distributions ($n$ = 31, within-subject design) under Placebo (**a**), L-dopa (**b**), and Haloperidol (**c**). Choices of the sub-optimal (20% reinforced) options in (**a**−**c**) are coded as negative RTs, whereas choices of the optimal (80% reinforced) options are coded as positive RTs. **d** Accuracy per drug condition (chance level is 0.5). **e** Total rewards earned per condition. **f** Median RT per drug condition. Pl – Placebo, L – L-dopa, H – Haloperidol. For boxplots, lines represent the median, the box covers the upper and lower quartiles, and the whiskers denote the range of datapoints falling within 1.5 times the interquartile range.

## Model comparison

We next compared three computational models (see methods section for details). As a reference, we first fit a null model ($DDM_O$) without a learning component (i.e. constant drift rates across trials). We then examined two reinforcement learning drift-diffusion models (RLDDMs) that included a linear mapping from Q-value differences to trial-wise drift rates[59,60,62] with either a single learning rate η (RLDDM1) or dual learning rates η for positive vs. negative prediction errors (RLDDM2). Model comparison was performed using the estimated log pointwise predictive density (-elpd)[65]. The RLDDM with dual learning rates outperformed both the single learning rate model and the $DDM_O$ model without learning, and this model ranking was replicated across all three drug conditions (Table 2).

## Posterior predictive checks

We next ran posterior predictive checks to examine the degree to which the best-fitting model (RLDDM2) accounted for key patterns in the data, in particular the increase in accuracy and the reduction in RTs over the course of learning. 10k datasets were simulated from each model's posterior distribution. Trials were binned into ten bins over the course of learning, and mean accuracies and RTs were computed per bin for both the observed data and across a 1k subset of simulated data sets, separately for each drug condition (Fig. 2).

As the $DDM_O$ predicts constant accuracies and RTs over trials, it cannot reproduce the observed increases in accuracy and reductions in RTs over the course of learning (Fig. 2). In contrast, both RLDDMs reproduced learning-related reductions in RTs (Fig. 2, left) and increases in accuracy (Fig. 2, right). However, RLDDM1 tended to underestimate both effects. In particular, it underpredicted accuracies (Fig. 2, right), and somewhat overpredicted RTs (Fig. 2, left). RLDDM2 provided a better account of these effects. In line with the model comparison, this pattern was observed in all drug conditions.

It was recently suggested that RLDDMs might fail to account for the full RT distributions[66]. Additional posterior predictive checks were, therefore, performed that examined predictions of 17th, 50th and 83rd percentiles of the RT distributions (Supplemental Fig. S2). Again, RLDDM2 reproduced these data well. RLDDM2 also reproduced individual subject RT distributions (Supplemental Figs. S3–S5), and accounted for the evolution of RTs over the course of learning in individual participants (Supplemental Figs. S6–S8), where for all participants the observed data fell within the 95% prediction interval of the simulated RTs.

## Analysis of drug effects on model parameters

We next analyzed the posterior distributions of the RLDDM2, focusing on a combined model in which the Placebo condition was modeled as the baseline, and drug effects were modeled as additive changes from the baseline, for each parameter (see methods section). Figure 3 (top row) depicts the posterior distributions of the group mean of each parameter under Placebo, whereas the mid- and center rows show effects of L-dopa and Haloperidol on each parameter (see also Table 3). Drug effects were quantified in three ways. First, we computed the posterior probabilities of each drug effect being <0 (see Table 3). This probability exceeded 97.5% for L-dopa and Haloperidol effects on boundary separation (decision threshold), and for Haloperidol effects on the negative learning rate. Next, we computed regions-of-practical equivalence[67] (ROPEs) in terms of ± 0.1 SD of each parameter under placebo (shaded areas in Fig. 3). Whereas the 95% highest posterior density interval (HDI) did not overlap with the ROPE or zero for the effect of Haloperidol on boundary separation (Fig. 3a), a small effect

## Table 1 | Descriptive statistics

| Condition | Accuracy | Total rewards | Median RT |
|---|---|---|---|
| Placebo | .871 (.163) | 40.839 (7.781) | .820 (.188) |
| L-dopa | .843 (.161) | 39.710 (6.394) | .788 (.143) |
| Haloperidol | .840 (.136) | 39.645 (5.395) | .789 (.126) |
| $BF_{incl}$ | <1.349 | <1.401 | <.943 |

[Mean (STD)] for model-agnostic performance measures and results from Bayesian repeated measures ANOVAs with linear and quadratic effects of working memory capacity and linear effects of body weight as covariates. Inclusion Bayes Factors ($BF_{incl}$) for all terms were <3, providing little credible evidence for effects of these terms on behavioral measures.

**Table 2 | Model comparison results, separately per drug condition**

| Model | η | Placebo<br>-elpd (SE) | *Rank* | L-dopa<br>-elpd (SE) | *Rank* | Haloperidol<br>-elpd (SE) | *Rank* |
|---|---|---|---|---|---|---|---|
| **DDM$_0$** | - | 296.5 (52.1) | 3 | 375.9 (52.8) | 3 | 480.2 (51.2) | 3 |
| **RLDDM 1** | Single | 178.7 (47.7) | 2 | 271.6 (50.0) | 2 | 412.7 (49.5) | 2 |
| **RLDDM 2** | Dual | 67.3 (48.5) | 1 | 195.4 (51.2) | 1 | 336.8 (50.7) | 1 |

We examined reinforcement learning drift-diffusion models (RLDDMs) with a single learning rate η (RLDDM 1) vs. separate learning rates for positive and negative prediction errors (RLDDM 2). We also included a null model without learning (DDM$_0$). Model comparison was conducted using the estimated log pointwise predictive density (-elpd)[65] where smaller values indicate a better fit.

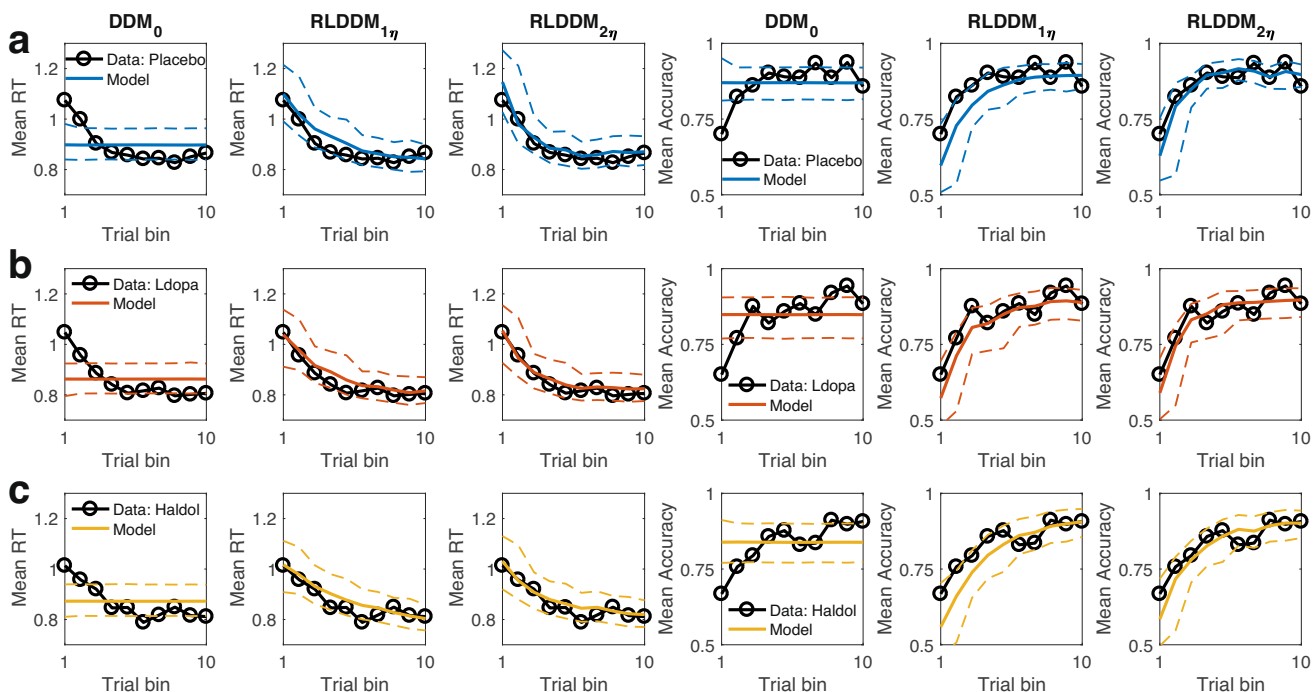

**Fig. 2 | Group-level posterior predictive checks.** DDM$_0$: Null model without reinforcement learning. RLDDM1η: Reinforcement learning drift-diffusion model with a single learning rate. RLDDM2η: Reinforcement learning drift-diffusion model with dual learning rates. Data and model simulations are shown for Placebo (**a**), L-dopa (**b**), and Haloperidol sessions (**c**). Left columns: observed mean group-level RTs over time (black lines) and model predicted RTs (solid colored lines: means, dashed lines: +/− 95% percentiles). Right columns: observed mean group-level accuracies over time (black lines) and model predicted accuracies (solid colored lines: means, dashed lines: +/− 95% percentiles).

size could not be ruled out with 95% confidence for L-dopa effects on boundary separation, and Haloperidol effects on the negative learning rate (Table 3 and Fig. 3a, e). Finally, Bayes Factors testing for directional effects are reported for each drug effect in Table 3.

The analysis of drug effects was repeated using a modeling scheme in which separate models were fit to the data from each drug condition. This reproduced the effects observed in the combined model (Supplemental Fig. S9, Supplemental Table S3). The only parameter showing drug effects was the boundary separation, which was reduced in both drug conditions, compared to Placebo.

As a control analysis, effects of working memory capacity (WMC, the first principal component of a principal component analysis across listening span, operation span and rotation span tasks, see Supplemental Fig. S10) and body weight were included in the hierarchical model as modulators of the drug effects. This revealed little credible evidence that any of the drug effects were reliably modulated by these covariates (see Supplemental Figs. S11–S13).

Based on the idea that basal ganglia output might convey an urgency signal[38], behavioral data were additionally fitted with an RLDDM with linearly collapsing bounds, as implemented in the HDDM toolbox[68–70]. In this model, both drugs were again associated with

reduced overall decision thresholds but little credible evidence was seen for an impact on the degree of threshold collapse (Supplemental Figs. S14 and S15).

Finally, to link the modeling results back to individual differences in behavior, we examined the degree to which drug effects on boundary separation were associated with differences in RTs and accuracy between conditions, focusing on the slowest third of trials. Prediction of ldopa effects on boundary separation using Bayesian linear regression (controlling for WMC, WMC$^2$ and body weight) revealed evidence for effects of RT differences (Fig. 4a, BF$_{incl}$ = 290.191) and accuracy differences (Fig. 4c, BF$_{incl}$ = 107.038). The corresponding analysis for haldol effects revealed only evidence for an effect of RT differences (Fig. 4b, BF$_{incl}$ > 10.000) but not accuracy differences (Fig. 4d, BF$_{incl}$ < 1).

**FMRI results**

FMRI analyses focused on a single a priori region of interest (ROI) based on two meta-analyses of value effects (see methods, including ventral striatum, ventromedial prefrontal cortex, posterior cingulate). Using parametric measures derived from our computational model, the first aim was to replicate the effects of model-derived values in

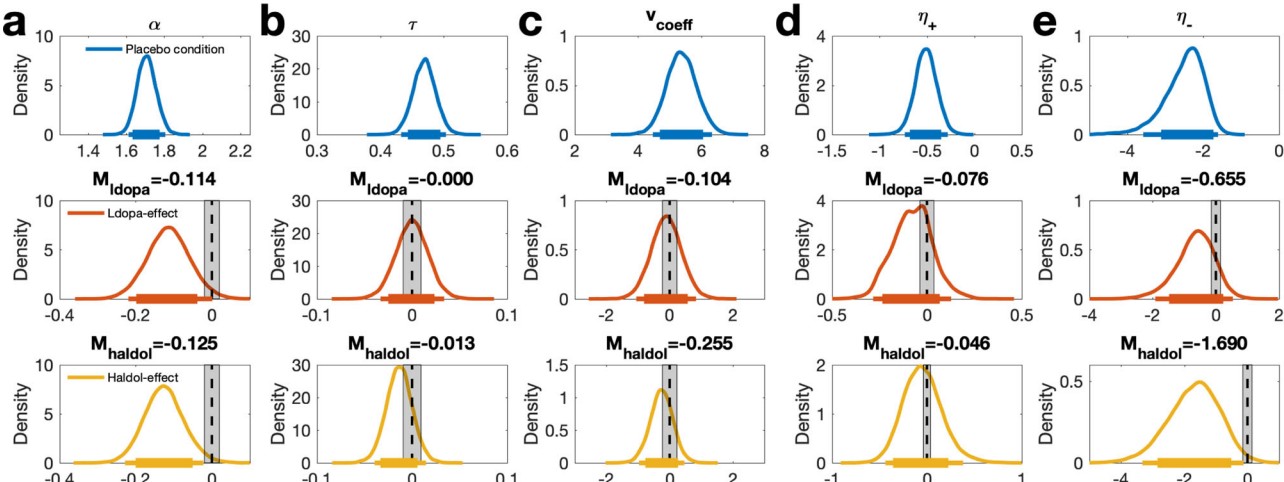

**Fig. 3 | Drug effects on RLDDM2 parameters. a** boundary separation ($\alpha$), **b** non-decision time ($\tau$), **c** value coefficient of the drift rate ($v_{coeff}$), **d** positive learning rate ($\eta_+$) in standard normal space, **e** negative learning rate ($\eta_-$) in standard normal space. Top row: Posterior distributions for each parameter under Placebo. Center row: Posterior distributions of L-dopa-effects on each parameter ($M_{ldopa}$ refers to the mean). Bottom row: Posterior distributions of Haloperidol-effects on each parameter ($M_{haldol}$ refers to the mean). Solid (thin) horizontal lines denote 85% (95%) highest posterior densities. Shaded areas denote Regions of Practical Equivalence (ROPEs)[67] ± 0.1 SD (based on the placebo condition).

**Table 3 | Drug effects on model parameters**

| RLDDM2 parameter | L-dopa effect | | | Haloperidol effect | | |
|---|---|---|---|---|---|---|
| | M [95% HDI] | P(effect < 0) | dBF (<0) | M [95% HDI] | P(effect < 0) | dBF (<0) |
| $\alpha$ | −.114 [−.219, .001] | .977 | 37.462 | −.125 [−.228, −.022] | .988 | 72.841 |
| $\tau$ | .000 [−.033, .034] | .502 | 1.093 | −.013 [−.039, .015] | .840 | 5.087 |
| $v_{coeff}$ | −.0104 [−1.057, .834] | .588 | 1.482 | −.255 [−.969, .467] | .762 | 3.181 |
| $\eta_+$ | −.076 [−.283, .127] | .776 | 3.290 | −.046 [−.438, .381] | .606 | 1.523 |
| $\eta_-$ | −.655 [−1.918, .538] | .870 | 6.192 | −1.69 [−3.323, −.125] | .987 | 78.686 |

Statistical testing was performed by directly examining the posterior distributions of group-level parameters. Mean posterior drug effects on model parameters ($M_{diff}$) relative to the Placebo condition, for L-dopa (left) and Haloperidol (right) and posterior probabilities that the given change in a parameter is <0, P(effect<0). Directional Bayes Factors (dBF) quantify the degree of evidence for a reduction in a parameter, relative to an increase. No correction for multiple comparisons was applied for these measures[99].

vmPFC/mOFC and prediction error in the ventral striatum. Main effects across drug conditions in the reward ROI replicated both effects (see Table 4, Fig. 4a: average Q-value, Fig. 4b: chosen – unchosen Q-value, Fig. 5: prediction error). Numerically, the effects of the average value were more pronounced than the chosen vs. unchosen value (Table 4, Fig. 5).

In the next step, we tested for drug effects on the same three effects via F-contrasts testing for the main effects of the drug. In none of the contrasts did we observe any effects that survived correction for multiple comparisons across the reward ROI. We also did not observe drug effects when running an FWE-corrected whole-brain analysis on these three effects. Finally, we tested for drug effects on stimulus-onset and feedback-onset-related effects, again using whole-brain FWE correction. No significant effects were observed.

To reproduce the analysis from Pessiglione et al.[17], positive and negative prediction error effects were extracted from bilateral ventral striatal regions that encoded model-derived prediction errors (see Fig. 6a and Table 4) in GLM1. Using GLM2 that included separate predictors for positive and negative prediction errors, the corresponding parameter estimates were extracted (Fig. 6b). While Pessiglione et al.[17] reported a greater contrast between positive and

negative striatal prediction errors under L-dopa compared to Haloperidol, this was not the case in our data. Bayesian ANOVAs with covariates of body weight, WMC, and WMC$^2$ and the factors prediction error (positive vs. negative) and drug (Placebo vs. L-dopa vs. Haloperidol) only revealed evidence for a prediction error effect (left ventral striatum, $BF_{incl} = 142.111$, right ventral striatum, $BF_{incl} = 6.138$, all other $BF_{incl} < 1.045$).

Given that no drug effects were observed in our a priori-defined region of interest, a final exploratory analysis was performed, testing for drug effects on average Q-value effects at an uncorrected threshold of $p < .0001$. This revealed higher effects under L-dopa and Haldol compared to placebo in the left anterior insula and the dorsal anterior cingulate/pre-SMA region (Supplemental Fig. S16 and Supplemental Table S4).

## Discussion

We used a stationary reinforcement learning task[17] in combination with a pharmacological manipulation of dopamine (DA) neurotransmission (Placebo, 150 mg L-dopa, 2 mg Haloperidol) and fMRI to address two core research questions. First, we examined a previously reported effect of L-dopa (vs. Haloperidol) on

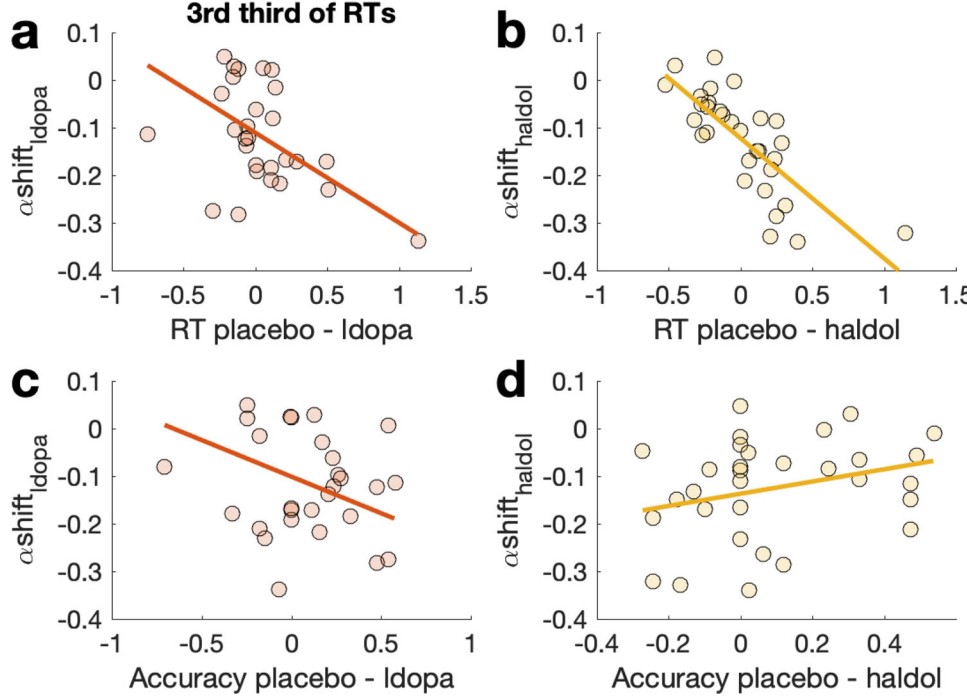

**Fig. 4 | Associations between model-based and model-agnostic drug effects.** Associations between drug effects (**a**, **c**: ldopa, **b**, **d**: haldol) on boundary separation α from the combined RLDDM2 (y-axis) and behavioral differences between conditions in the slowest third of trials. A, **b**: RT difference between placebo and drug. **c**, **d**: difference in arcsine-sqrt transformed accuracy between placebo and drug.

reinforcement learning and striatal prediction error signaling[17]. These effects were not observed - there was no credible evidence for improved learning under L-dopa. Perhaps unsurprisingly, given the lack of a behavioral effect, we also did not observe credible evidence for drug effects on positive vs. negative prediction error coding in the ventral striatum, again contrasting with Pessiglione et al.[17], but in line with recent studies using different reinforcement learning tasks[19,20]. Potential reasons are discussed further below. Second, we leveraged recently developed combined reinforcement learning drift diffusion models (RLDDMs)[59–63] to directly test a recently proposed computational account of dopamine in regulating decision thresholds during action selection[38]. In line with this

**Table 4 | Replication analysis for three model-derived measures (main effects across drug conditions), average Q-value across options, chosen – unchosen Q-value, and model-derived prediction error**

| Contrast / *Region* | Coordinates | Peak T-value | $p(FWE)_{SVC}$ |
|---|---|---|---|
| Average Q-value | | | |
| *vmPFC* | −6 48 −2 | 5.06 | .002 |
| Chosen-unchosen value | | | |
| *vmPFC / mOFC* | 6 24 −12 | 4.09 | .033 |
| Reward prediction error | | | |
| *Left ventral striatum* | −9 9 −9 | 5.95 | <.001 |
| *Right ventral striatum* | 12 9 −10 | 4.92 | .003 |
| *vmPFC / mOFC* | −6 32 −15 | 4.19 | .027 |
| *vmPFC* | −6 54 3 | 4.08 | .036 |

Small volume correction for multiple comparisons (SVC) used an a priori region of interest mask across two meta-analyses of reward value effects[97,98] (see methods section).

account, computational modeling revealed reduced decision thresholds under both L-dopa and Haloperidol, compared to Placebo. The latter effect is consistent with the present Haloperidol dosage of 2 mg increasing (rather than decreasing) striatal DA by blocking presynaptic autoreceptors (see discussion below).

We aimed to replicate the core behavioral finding from Pessiglione et al.[17], but for practical reasons deviated from their experimental design in a number of ways. First, the drug dosages in the two projects differed slightly – we used 150 mg of L-dopa (compared to 100 mg in Pessiglione et al.) and 2 mg of Haloperidol (compared to 1 mg in Pessiglione et al.). The L-Dopa dosage was chosen to keep drug dosages comparable to related work[71,72]. The Haloperidol dosage was selected to keep drug dosages comparable to planned studies investigating other effects[73,74], and prior to completion of our human work suggesting that 2 mg Haloperidol might elicit effects more compatible with an increase in DA transmission[33,75]. Second, we only included the gain condition, because here, the primary behavioral effect was observed. Although we doubled the number of gain trials (i.e. we used two pairs of symbols), our task version was likely still easier than the one used by Pessiglione et al.[17], who used three pairs of symbols. Furthermore, this isolation of the gain condition might have affected drug effects on learning. The reason is that, in the present task version, the mean initial reward expectation was positive (there were only gains or reward omissions), yielding both positive and negative prediction errors during initial learning. In contrast, in Pessiglione et al., the mean initial reward expectation was zero (there were equally many gain and loss trials), yielding only positive prediction errors in the gain condition during initial learning. This might have masked drug effects. Third, we increased the sample size to $n = 31$, and applied a within-subjects design as opposed to a between-subjects design. Although this might have induced learning across sessions, we observed no credible evidence for performance changes over time, and restricting the analysis to the first session likewise revealed no drug effects. The most straightforward replication attempt of the behavioral effect, a comparison of total rewards obtained between the L-dopa and Haloperidol

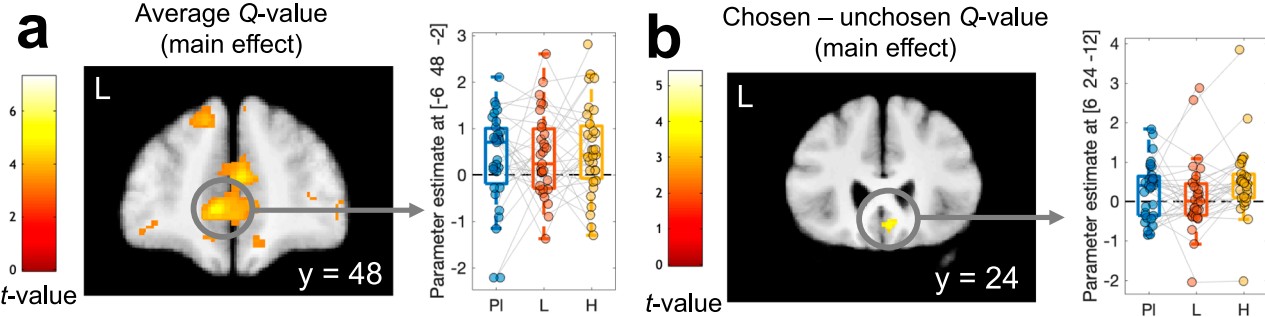

**Fig. 5 | Model-based FMRI results for *Q*-values.** (*n* = 31, flexible factorial model in SPM12 with within-subjects factor of drug condition) for model-derived average Q-value (**a**, contrast testing for an effect > 0 across all drug conditions) and chosen − unchosen Q-value (**b**, contrast testing for an effect > 0 across all drug conditions). Correction for multiple comparisons was performed using a meta-analysis-based region-of-interest mask (see Table 4 and methods section). Maps are thresholded at *p* < .001 *uncorrected* for display purposes, and projected on the group mean T1 scan. Pl – Placebo, L – Levodopa, H – Haloperidol. For boxplots, lines represent the median, the box covers the upper and lower quartiles, and the whiskers denote the range of datapoints falling within 1.5 times the interquartile range.

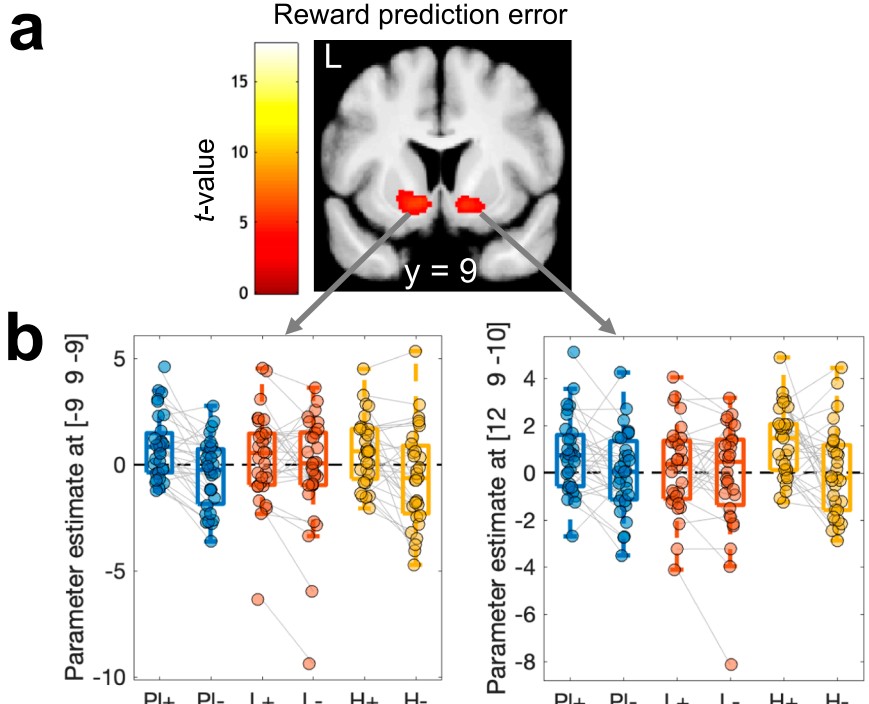

**Fig. 6 | Model-based FMRI results for prediction error coding (n = 31).** Striatal regions coding for model-based prediction errors were identified by computing a main effect of prediction error across all drug conditions using GLM1 (**a**, flexible factorial model in SPM12 with within-subjects factor of drug condition). Correction for multiple comparisons was performed using a meta-analysis-based region-of-interest mask (see Table 4 and methods section). To reproduce the analysis of Pessiglione et al.[17], we then extracted parameter estimates at left and right striatal peak voxels (see Table 5) from GLM2 (flexible factorial model in SPM12 with within-subjects factors of drug condition and prediction error sign) to obtain parameter estimates for positive (+) and negative (−) prediction errors, separately for each drug condition (**b**). The map in (**a**) is thresholded at *p* < .001 *uncorrected* for display purposes, and projected onto the group mean T1 scan. Pl – Placebo, L – Levodopa, H – Haloperidol; +: positive prediction error, −: negative prediction error. For boxplots, lines represent the median, the box covers the upper and lower quartiles, and the whiskers denote the range of datapoints falling within 1.5 times the interquartile range.

conditions via a frequentist paired t-test, yielded no credible evidence for an effect.

We consider the isolation of the gain condition to be the most likely reason for the lack of replication, although other accounts are possible. For example, differences in dosages might account for the lack of a drug effect on performance, but we consider this unlikely, for several reasons. First, 150 mg of L-dopa yielded positive behavioral effects in a range of other studies[15]. Second, 2 mg of Haloperidol is a dosage that likely predominantly affects presynaptic autoreceptors and thus likely increases (rather than decreases) striatal DA release[8,28–33,75]. Note that this generally questions a

common interpretation of a downregulation of striatal DA using even lower dosages of Haloperidol[17,71]. Nonetheless, even if one would argue in favor of a DA-downregulation account of 1 mg or 2 mg of haloperidol (which we consider unlikely), in this case one would, if anything, expect more pronounced effects of our 2 mg dosage compared to Pessiglione et al.'s 1 mg dosage. Yet, inconsistent with this idea, no credible evidence for drug effects on performance was observed. Also, the relatively small sample size of the original study might have led to an increased risk of false positives. Finally, there are substantial individual differences with respect to pharmacological effects of DA drugs[1] and it is possible

that such individual differences might have also contributed to the inconsistencies between studies.

Extending previous work, we applied RLDDMs to directly examine the effects of DA on the dynamics underlying action selection. The model comparison revealed that the data were best accounted for by an RLDDM with separate learning rates for positive and negative prediction errors (RLDDM2), and the model ranking was replicated in each drug condition. Good parameter recovery for RLDDMs was recently confirmed for data from the task used here[58]. Posterior predictive checks likewise confirmed that RLDDM2 provided a good account of learning-related increases in accuracy and decreases in RT. Likewise, additional checks across different percentiles of the RT distributions revealed that RLDDM2 provided a good account of the data (Supplemental Fig. S2). Although RLDDMs have been suggested to provide a comparatively poor account of full RT distributions[66], this was not observed here. Finally, we confirmed that RLDDM2 also provided a good account of individual-participant RT distributions in each drug condition (Supplemental Figs. S3–S5), consistent with prior work[75,76]. The model also reproduced learning-related changes in RTs over the course of learning in individual participants (Supplemental Figs. S6–S8). In contrast to earlier work[54,61,63,75,77], models with a non-linear mapping from value differences to trial-wise drift rates failed to converge (potentially due to lower trial numbers), and we, therefore, focused on a model with a linear linkage function[59,62].

RLDDM2 revealed that, compared to Placebo, both L-dopa and Haloperidol reduced decision-thresholds. This effect was observed both in the combined model across all drug conditions (Fig. 3), in a control analysis using separate models for each drug condition (Supplemental Fig. S9, Supplemental Table S3), and when stable bounds were replaced with a collapsing bounds mechanism (Supplemental Figs. S14 and S15). Although we did not observe strong evidence for drug effects on model-agnostic behavioral measures, the overall pattern of behavioral results is nonetheless consistent with the observed reduction in decision thresholds: numerically, under both L-dopa and Haloperidol, accuracy was lower, and median RTs were faster. Furthermore, individual differences in drug-induced decision threshold reductions accounted for individual differences in RT differences between conditions. This argues against the idea that e.g. excessive shrinkage of parameters modeling drug effects might have driven drug effects in the combined RLDDM. Notably, for learning-related effects, previously reported drug effects did not replicate (see discussion above), whereas, for action selection, consistent reductions in decision thresholds were observed. Learning-related effects may be more affected by the omission of the loss condition than action-selection effects.

We observed similar reductions in decision thresholds following L-dopa and Haloperidol administration, two very different dopaminergic agents. By increasing substrate availability, L-dopa is assumed to generally increase DA availability. In contrast, Haloperidol is a D2 receptor antagonist that likely exhibits dose-dependent effects on striatal DA release. Specifically, lower dosages are thought to predominantly affect presynaptic inhibitory autoreceptors[27], thereby increasing DA release[8,28–33]. Along similar lines, other D2 antagonists potentiate striatal effects at lower dosages[25], and attenuate them at higher dosages[23]. A Haloperidol dosage of 2 mg has been shown to substantially upregulate human striatal responses[33]. Notably, however, the dose-dependency of Haloperidol might additionally be region-dependent[24], which further complicates the interpretation of the effects.

Our finding of reduced thresholds following pharmacological increases in DA also resonates with some findings in Parkinson's Disease (PD). PD patients when tested ON vs. OFF their DA medication sometimes show increased speed but impaired accuracy[78], and a reduced ability to suppress premature actions[79], consistent with reduced decision thresholds. However, other studies did not find

evidence for reduced decision thresholds in PD patients ON vs. OFF medication during perceptual decision-making[80].

This null-effect in perceptual decision-making in PD patients might also point to a more general pattern – the specific domain of the decision problem might determine the degree to which DA contributes to threshold adjustments. During perceptual decision-making, Bromocriptine, a DA agonist, did likewise not modulate decision thresholds[56]. As noted by the authors, this might also be due to dose-dependent presynaptic effects of Bromocriptine, which might have resulted in a net reduction in DA transmission[56]. But another possibility is that DA might specifically contribute to threshold adjustments in value-based decision settings[54]. This idea resonates with the role of dopamine in regulating response vigor[34–37]. In some of these accounts, DA is thought to signal whether increases in cognitive or physical effort (in some cases equivalent to increases in response rate) are worthwhile[37,42,81,82]. Adjustments in decision thresholds would constitute a simple computational mechanism to accomplish this. Reductions in decision thresholds during decision-making under high DA and increases in response vigor in effort-related tasks might thus both serve the same purpose of (potentially) increasing the reward rate. Such an account would predict that action selection in the context of high reward options (where the cues themselves likely elicit phasic DA release during the choice phase[13]) would likewise lead to a downregulation of decision thresholds. Notably, this is exactly what has been observed in recent work[63,83].

Increased DA is thought to shift the activation balance between striatal *go* and *no go* pathways towards the *go* pathway[13,14,39], thereby facilitating action execution vs. inhibition. In some models, separate *go* and *no go* action weights are modeled for each action[13,14]. An increase in DA during choice (be it pharmacological, as in the present study, or incentive-based[63,83]) would then for each action boost the contrast of *go* and *no go* action weights[13,14], leading to a general increase in the probability of action initiation. In a sequential sampling modeling framework, this could be captured by a reduction of decision thresholds[14] (reduced boundary separation). Conceptually, this is related to the idea that DA signals the (subjective) precision of beliefs[40–42]. If one's belief in the precision of action weights is increased under high DA, this would naturally lead to accepting less evidence prior to committing to a decision. However, given the lack of drug effects on striatal activation, our imaging results remain agnostic with respect to a modulation of striatal computations. Furthermore, given the different mechanisms of action, there remains a possibility that both drugs might have affected decision thresholds via different routes.

Haloperidol also reduced the negative learning rate, although this effect was only observed in the combined model. This might be linked to a decrease in the *no go* pathway, and resonates with some earlier findings[8,23]. However, another study did not observe changes in learning rate under Haloperidol, but here only a single learning rate was estimated, and no RLDDM modeling scheme was applied[24].

Exploratory analyses implicated a circuit involving anterior insula and ACC / preSMA in the drug-induced decision threshold modulation (Supplemental Figure S16), such that average value effects in these regions were higher under L-dopa and Haloperidol compared to placebo. Caution is warranted when interpreting these condition-dependent differences in parametric effects[84], in particular, given their exploratory nature. Nonetheless, the dorsal anterior cingulate / pre-SMA region has previously been implicated in control adjustments[85], including decision threshold modulation[45,51,86,87], and is densely interconnected with subcortical dopaminergic circuits[9]. A contribution of this circuit to the observed decision threshold adjustments under elevated DA would therefore be in line with these previous findings, but further confirmatory evidence is required.

The present study has a number of limitations that need to be acknowledged. First, we only tested male participants, limiting the

generalizability of our findings. Second, the present sample size of n = 31 is still too low to comprehensively examine potential non-linear baseline-dependent drug effects[1,19,88]. Third, the task was performed following completion of a separate learning task[19], i.e. about 60 min post ingestion of L-dopa. L-dopa reaches peak plasma levels around 30-60 min post ingestion, with a plasma half-life of about 60-90 min[89,90]. The present task was therefore likely performed past the time point of peak L-dopa plasma levels, but likely before the plasma half-life was reached. Although it is possible that this timing may have contributed to the lack of L-dopa effects on learning and neural prediction error signaling, we consider this unlikely, for several reasons. First, in the learning task performed directly prior to the task reported here[19], as well as in other studies[20], L-dopa likewise failed to modulate striatal prediction error signaling. Second, the robust L-dopa effect on boundary separation argues against a drug timing account to explain the lack of modulation of the prediction error response. Finally, under some conditions, dopaminergic manipulations can also affect evidence accumulaton[75,91], such as the impact of benefits and costs on choice[92]. Further work is required to determine the degree to which these effects depend on the task or the specifics of the pharmacological manipulation.

To conclude, we observed no credible evidence for a beneficial effect of L-dopa (vs. Haloperidol) on reinforcement learning in a reward context, as well as the proposed mechanistic account of an enhanced striatal prediction error response mediating this effect. Differences in experimental design between studies likely account at least in part for this. In contrast, across a variety of modeling schemes, RLDDMs revealed robust reductions in decision thresholds under both L-dopa and Haloperidol. This provides evidence for a recently proposed computational account of the role of DA in action selection[38,39], and is consistent with both drugs boosting action-specific activation contrasts between striatal *go* and *no go* pathways during choice[13,14], which might have impacted subsequent action selection mechanisms in ACC / pre SMA. Such a threshold modulation account of DA can potentially bridge circuit-level accounts of action selection in the basal ganglia[13,39] with a proposed role of dopamine in regulating response vigour[34–37].

## Methods
### General procedure
All study procedures were approved by the local ethics committee (Hamburg Board of Physicians) and participants provided informed written consent prior to participation. Data were obtained in the context of a larger multi-day combined pharmacological fMRI study[19] with four testing sessions, performed on separate days. Day 1 consisted of a behavioral testing session, during which working memory (operation span, listening span, rotation span, see Chakroun et al. (2020) for details[19]) and questionnaire data were obtained. On days 2 – 4 (each performed exactly 1 week apart), healthy self-identified male participants (*n* = 31, age 19-35, M = 26.85, SD = 4.01) received either Placebo, L-dopa or Haloperidol (in counterbalanced order, see below for details) and then performed two tasks while brain activity was measured using fMRI. The first task was a restless four-armed bandit task measuring exploration/exploitation behavior. Data from this task have been reported elsewhere[19]. Following a short break, participants then completed the stationary reinforcement learning task based on a previous study[17], which is reported here.

### Drug administration
Participants performed three fMRI sessions (in counterbalanced order) under three drug conditions: Placebo, L-dopa (150 mg), and Haloperidol (2 mg). They arrived in the lab 2.5 h prior to the commencement of fMRI scanning. Upon arrival, they received a first pill containing either 2 mg haloperidol or placebo (maize starch). Exactly two hours later, participants received a second pill containing either

Madopar (150mg L-dopa + 37.5 mg benserazide) or Placebo. That is, during the Placebo session, participants received maize starch/maize starch. During the Haloperidol session, they received Haloperidol/ maize starch, and during the L-dopa session, they received maize starch / L-dopa.

Half an hour after ingesting the second pill, participants entered the fMRI scanner, where they first performed a restless four-armed bandit task reported elsewhere[19]. Following a short break inside the scanner, they then performed 60 trials of the stationary reinforcement learning task reported here.

Physiological parameters, well-being, potential side effects, and mood were assessed throughout each testing session[19]. No participant reported any side effects.

### Reinforcement learning task
During each fMRI session, participants performed a simple stationary reinforcement learning task (see Supplementary Figure S17) based on a previous study[17]. The task involved two pairs of fractal images (n = 30 trials per pair). Per pair, one stimulus was associated with a reinforcement rate of 80% ("optimal" stimulus) whereas the other was associated with a reinforcement rate of 20% ("suboptimal" stimulus). On each trial, options were randomly assigned to the left/right side of the screen, and trials from the two options pairs were presented in randomized order. Following the presentation of the two options, participants had a maximum of 3 seconds to log their selection via a button press. Participants received binary feedback, either in the form of the display of a 1€ coin (*reward* feedback) or as a crossed 1€ coin (*no reward* feedback). Jitters of variable duration (2-6 sec, uniformly distributed) were included following presentation of the selection feedback and following the presentation of the reward feedback. Prior to scanning, participants performed a short practice version of the task with a different set of stimuli in order to familiarize themselves with the procedure.

### Statistical analyses
Drug effects on model-free performance measures and fMRI parameter estimates extracted at specific peaks were analyzed via Bayesian repeated measures ANOVAs using the JASP software package (Version 0.16.3)[64]. The normality of residuals was verified via Q-Q-plots, which showed no credible evidence for deviations from normality (see Supplemental Fig. S18).

### Q-learning model
We applied a simple Q-learning model[4] to formally model the learning process. Here, participants are assumed to update the value (Q-value, Eq. 1) of the chosen option based on the reward prediction error that is computed on each trial as the difference between the obtained reward and the expected reward. The degree of updating is regulated by the learning rate $\eta$ (Eq. 2):

$$Q_{chosen,t+1} = Q_{chosen,t} + \eta * \delta_t \tag{1}$$

$$\delta_t = r_t - Q_{chosen,t} \tag{2}$$

Q-values of unchosen actions remain unchanged. All Q-values were initialized with values of .5. As learning from positive and negative reinforcement is thought to depend on distinct striatal circuits[7,93], we compared models with a single learning rate ($\eta$) and dual learning rates ($\eta_+$, $\eta_-$) for positive vs. negative prediction errors. Learning rates were estimated in standard normal space [−6, 6] and back-transformed to the interval [0, 1] via the inverse cumulative normal distribution function.

## Reinforcement learning drift-diffusion models (RLDDMs)

We used combined reinforcement learning drift-diffusion models (RLDDMs)[59,60,63] to link participants' choices and response times (RTs) to the learning model. In the drift-diffusion model, choices arise from a noisy evidence accumulation process[43,44] that terminates as soon as the accumulated evidence exceeds a threshold, i.e. crosses one of two response boundaries. The upper response boundary was defined as the selection of the optimal (80% reinforced) stimulus, whereas the lower response boundary was defined as the selection of the sub-optimal (20% reinforced) stimulus. RTs for choices of the suboptimal option where multiplied by -1 prior to model estimation, and we discarded for each participant the fastest 5% of trials in order to ensure that implausibly fast trials do not exert an undue influence on the modeled RT distributions. For comparison with the RLDDMs, we first examined a null model without learning (DDM$_0$). Here, the RT on each trial $t$ is distributed according to the Wiener First Passage Time (*wfpt*):

$$RT_t \sim wfpt(\alpha, \tau, z, \nu) \tag{3}$$

The boundary separation parameter $\alpha$ controls the speed-accuracy trade-off (decision threshold), such that smaller values of $\alpha$ lead to faster but less accurate responses. The drift rate $\nu$ reflects the quality of the evidence, such that greater values of $\nu$ give rise to more accurate and faster responses. Note that in the DDM$_0$, $\nu$ is constant and unaffected by learning. The non-decision time $\tau$ models RT components related to motor and/or perceptual processing that are unrelated to the evidence accumulation process. The starting point parameter $z$ models a bias towards one of the response boundaries. Following earlier work[54,59,61-63] and based on the assumption that participants do not have an a priori bias for optimal vs. suboptimal options prior to learning, $z$ was fixed to .5. Because a bias might nonetheless develop over the course of learning[59], we also explored a model with variable starting point. However, this model failed to converge (maximum $\hat{R} > 7$), even when restricting the analysis to the data from the placebo condition, and when sampling was increased by a factor of ten. This model was therefore not explored further.

Following earlier work[58,59,62,63] we then incorporated the learning process (Eqs. 1 and 2) into the DDM by setting trial-wise drift rates to be proportional to the difference in Q-values between the optimal and suboptimal options via a simple linear linkage function:

$$\nu_t = \nu_{coeff} * \left( Q_{optimal,t} - Q_{suboptimal,t} \right) \tag{4}$$

In the RLDDM, the RT on trial $t$ then depends on this trial-wise drift rate:

$$RT_t \sim wfpt(\alpha, \tau, z, \nu_t) \tag{5}$$

Here, $\nu_{coeff}$ models the degree to which trial-wise drift rates scale with the value difference between options. In this model, increasing Q-value differences lead to both increased accuracy and faster RTs. Conversely, when Q-values are similar, choices will be both less accurate and slower.

Previous work also examined non-linear linkage functions[63,75,77]. However, in the present data, models with non-linear linkage functions failed to converge. We therefore focused on Eq. 4, as originally proposed by Pedersen and colleagues[59] and successfully applied in other learning tasks[62]. This simpler model nonetheless reproduced key patterns in the data, in particular the increase in accuracy over trials, the decrease in RTs over trials, as well as individual-participant RT distributions.

In an earlier report[58], we additionally examined models in which $\alpha$ and/or $\tau$ were allowed to vary over the course of the experiment according to a power function. This modification of the RLDDM was motivated by the observation that, in participants suffering from disordered gambling, reinforcement learning only accounted for a relatively small part of the observed RT reductions over time, such that an additional mechanism was required to account for these data. In contrast, in the present study, posterior predictive checks revealed that this was not the case in any drug condition. Therefore, we refrained from examining these more complex models, instead focusing on models with constant $\alpha$ and $\tau$.

## Hierarchical Bayesian models

Models were fit to all trials from all participants using a hierarchical Bayesian modeling approach with separate group-level Gaussian distributions for all parameters. We ran two types of models. First, we fit separate hierarchical models to the data from each drug condition and compared the model ranking across conditions. After confirming that the model ranking was unaffected by the drug, we set up a combined hierarchical model in which parameters in the Placebo condition were modeled as the baseline, and deviations from placebo under L-dopa and Haloperidol were modeled for each parameter using additive shift parameters with Gaussian priors centered at 0. Posterior distributions were estimated using Markov Chain Monte Carlo via JAGS (Version 4.3)[94] using the Wiener module[95], in combination with Matlab (The MathWorks) and the *matjags* interface (https://github.com/msteyvers/matjags). For group-level parameter means in the placebo condition in the combined model, as well as in the separate models per drug condition, we used uniform priors defined over numerically plausible parameter ranges (see Supplemental Table S1). For drug-effects in the combined model we used Gaussian priors centered at zero (see Supplemental Table S1). For group-level standard deviations, we used uniform priors over numerically plausible ranges (see Supplemental Table S2).

For each model, we ran 2 chains with a burn-in period of 100k samples and thinning factor of 2. 10k additional samples were then retained for analysis. Chain convergence was assessed by examining the Gelman-Rubinstein convergence diagnostic $\hat{R}$, and values of $1 \le \hat{R} \le 1.01$ were considered as acceptable for all group-level and individual-participant parameters. Relative model comparison was performed via the estimated log pointwise predictive density (*elpd*)[65], an approximation of the leave-one-out cross-validation accuracy of a model.

For comparison, RLDDMs with both stable and collapsing bounds were additionally fitted using the HDDM toolbox[68,69] (version 0.9.8) via the HDDMRL and HDDMnnRL functions. Note that for these analyses, only models with a single learning rate were examined, since dual learning rates are not implemented in HDDMnnRL.

### Analysis of posterior distributions

Posterior distributions were analyzed in three ways. Posterior probabilities for drug effects <0 (i.e. the proportion of the posterior distributions falling below zero) as well as posterior means and 95% highest posterior density intervals are reported. Additionally, for drug effects, Bayes Factors testing for directional effects (dBF) are reported that quantify the degree of evidence for a reduction in a given parameter relative to the evidence for an increase.

### Parameter recovery

We have previously reported parameter recovery analyses for RLDDMs in the context of the same task studied here[58]. These analyses revealed good parameter recovery for both individual-level and group-level parameters using the same Bayesian estimation methods applied here.

### Posterior predictive checks

To ensure that the best-fitting models captured key aspects of the data, in particular the increases in accuracy and reductions in RTs over the course of learning, we performed posterior predictive checks as follows. We simulated 10k data sets from each model's posterior

distribution, separately for each drug condition. For group-level posterior predictive checks, per drug condition, we selected 1k simulated data sets, and for each simulated participant split trials into ten-time bins. For each bin, we then calculated group average accuracy and RT as well as means and 2.5% and 97.5% percentiles over the simulated data. Simulated data were then overlaid over the observed data. For individual participant posterior predictive checks, per drug condition, we again selected 1k simulated data sets. In first step, we simply overlaid observed and simulated RT distributions, separately for each condition. To examine how the models accounted for learning-related RT changes in individual participants, we then split both simulated and observed trials into five time bins. For each bin, we again calculated individual-participant means as well as means and 2.5% and 97.5% percentiles over the simulated data.

## FMRI data acquisition

MRI data were collected on a Siemens Trio 3 T system using a 32-channel head coil. In each session, participants performed a single run of 60 trials (following a short break after completion of our previously reported task[19]), yielding a total of 180 trials per participant. Each volume consisted of 40 slices (2 × 2 x 2 mm in-plane resolution and 1-mm gap, repetition time = 2.47 s, echo time 26 ms). We tilted volumes by 30° from the anterior and posterior commissures connection line to reduce signal dropout in the ventromedial prefrontal cortex and medial orbitofrontal cortex[96]. Participants viewed the screen via a head-coil-mounted mirror, and logged their responses via the index and middle finger of their dominant hand using an MRI-compatible button box. High-resolution T1-weighted structural images were obtained following the completion of the cognitive tasks.

## FMRI preprocessing

All preprocessing and statistical analyses of the imaging data were performed using SPM12 (Wellcome Department of Cognitive Neurology, London, United Kingdom). Volumes were first realigned and unwarped to account for head movement and distortion during scanning. Second, slice time correction to the onset of the middle slice was performed to account for the shifted acquisition time of slices within a volume. Third, structural images were co-registered with the functional images. Finally, all images were smoothed (8 mm FWHM) and normalized to MNI space using the DARTEL tools included in SPM12 and the VBM8 template.

## FMRI statistical analysis

Error trials were defined as trials where no response was made within the 3 sec response window, or trials that were excluded from the computational modeling during RT-based trial filtering (see above, for each participant, the fastest 5% of trials were excluded). We then set up first-level general linear models (GLMs) for each participant and drug condition. We used GLM1 and GLM3 for all main analyses, and GLM2 to reproduce a key analysis from Pessiglione et al.[17].

GLM1 included the following regressors:
1. onset of the decision option presentation
2. onset of the decision option presentation modulated by chosen – unchosen value
3. onset of the decision option presentation modulated by (chosen – unchosen value) squared
4. onset of the feedback presentation
5. onset of the feedback presentation modulated by model-derived prediction error
6. onset of the decision option presentation for error trials and
7. onset of the feedback presentation for error trials.

To separate out the effects of positive vs. negative prediction error coding, as done in Pessiglione et al.[17], a second first-level GLM was set up. GLM2 included the following regressors:

1. onset of the decision option presentation
2. onset of the decision option presentation modulated by chosen – unchosen value
3. onset of the decision option presentation modulated by (chosen – unchosen value) squared
4. onset of the feedback for positive prediction errors
5. onset of the feedback for negative prediction errors
6. onset of the decision option presentation for error trials and
7. onset of the feedback presentation for error trials.

GLM3 explored the effects of average Q-values during the choice phase. To this end, for regressors 2 and 3 in GLM1, the chosen – unchosen value was replaced with the mean Q-values across options.

To reproduce the analysis depicted in Fig. 3 of Pessiglione et al.[17], striatal activation peaks encoding model-derived prediction errors were first identified using contrast 5 in GLM1 (see below for the metanalysis-based region of interest mask that was applied). Parameter estimates for positive and negative prediction errors at these peak voxels were then extracted from GLM2.

Following Pessiglione et al. (2006), values and prediction errors were calculated using the condition-specific group-mean learning rates of the best-fitting model RLDDM2 (see Supplemental Table S5). All parametric regressors were z-scored within participants prior to entering the first level model[84]. Single-participant contrast estimates were then taken to a second-level random effects analysis using the flexible factorial model as implemented in SPM12 with a single within-subject factor of drug condition.

Analyses focused on reward-related circuits using a region of interest (ROI) mask provided by the Rangel Lab (https://www.rnl.caltech.edu/resources/index.html) which is based on two meta-analyses[97,98]. This mask was used for small-volume correction for our analyses of average Q-value, chosen – unchosen Q-value, and prediction error, as well as for testing for drug effects. For all effects, we plot single-participant contrast estimates extracted from group-level activation peaks.

## Reporting summary

Further information on research design is available in the Nature Portfolio Reporting Summary linked to this article.

## Data availability

Behavioral data generated in this study as well as fitted models have been deposited on OSF (https://osf.io/8vzgh/). Unprocessed fMRI data are protected and are not available due to data privacy laws. The processed 2nd-level fMRI data are deposited on OSF (https://osf.io/8vzgh/).

## Code availability

Model code is available on OSF (https://osf.io/8vzgh/).

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

## Acknowledgements

This work was funded by Deutsche Forschungsgemeinschaft (PE 1627/5-1 to J.P.). We thank the MRI staff at the Institute for Systems Neuroscience, UKE, Hamburg, for support during data acquisition, and Mathias Pessiglione for helpful discussions and comments on an earlier version of the manuscript. T. v. E. was supported by Deutsche Forschungsgemeinschaft (Project ID 431549029, CRC 1451).

## Author contributions

J.P., K.C., and A.W. designed the study. K.C. and F.G. acquired the data. J.P., A.W., and B.W. analyzed the data. J.P. wrote the paper. K.C., A.W., D.M., B.W., T.S., F.G., and T. vE. provided revisions and theoretical input. J.P. acquired the funding and supervised the project.

## Funding

## Competing interests

The authors declare no competing interests.
