## [Peer Review File · Nature Communications]

Dopamine regulates decision thresholds in human reinforcement learningREVIEWER COMMENTS

Reviewer #1 (Remarks to the Author):

In the current manuscript, Chakroun et al. present a study investigating the effect of L-Dopa and Haloperidol on a reinforcement learning task on 31 human participants in a blinded within-participants placebo-controlled study. fMRI data was collected while participants performed the task. Behavioural data was analysed using the reinforcement learning drift diffusion model.

The analysis of behavioural and BOLD data did not replicate effects of increased reward prediction errors from L-Dopa vs. Haloperidol (Pessiglione et al., 06). Instead, the authors found that administration of both L-Dopa and Haloperidol lead to a decrease in decision threshold, a parameter from the RLDDM measuring how decision makers balance the speed-accuracy tradeoff.

I found this to be a very interesting manuscript focusing on an important topic, in which the analysis was performed soundly and the behavioural modeling steps were well described. However, I do have some questions and comments. My points are mainly focused on the computational model, and less so on the neurobiology.

Replication

The authors spend some time describing the study as a replication of Pessiglione et al., 06. However, the lack of replication is mainly interpreted as resulting from a difference in the dosage prescribed and the task design. But if that's the case, then why do you find effects for the decision threshold? It also makes the reader very curious to see whether the effect would indeed replicate in a more similar replication.

Aims of paper

Related to the previous point, it is unclear whether the main goal was to replicate, or to discover whether dopamine is associated with reductions in decision threshold. From a critical perspective it could look like the discovery of a reduced decision threshold was used to change the focus from a replication study to a study investigating the effect of DA on cognitive mechanisms underlying learning and decision making. I don't think either approach is wrong, but that the initial goal of the study should be more clear.

Model

I thought the modeling steps and results were very well described. The following comments are just thoughts on additional approaches that could be interesting.

As described in Berke 18, the role of DA is proposed to influence decision threshold. But specifically, they propose that the effect could be an urgency signal or a reduction in threshold during the decision (collapsing bound model). This hypothesis could be tested using a newly developed method to incorporate other sequential sampling models than the DDM. It would be interesting to see whether the effect of L-Dopa and Haloperidol are better described as an increased collapse in threshold compared to a general reduction. A tutorial on applying such a model for an RL-task is described here: https://hddm.readthedocs.io/en/latest/demo_HDDMnnRL.html

It seems to me that theories of the effect of DA on choice could also be interpreted as having an influence on the rate of evidence accumulation through boosting go vs. no-go actions weights (e.g., Collins & Frank, 14), instead of having an effect on decision threshold. Could you provide more justification as to why boosting action weights should influence decision threshold and not drift rate?

I found the effect of Haloperidol on learning rate for negative feedback to be interesting, as it seems to relate to a decrease in the no-go pathway. Could the authors provide some reflections on this effect?

I would be interested to see whether you find any associations in the BOLD data related to the effect on decision threshold.

Minor

Line 333: could be confusing for some to use model-based (i.e. as in model-based vs. model-free). Maybe instead use model-derived?

Line 355: why did you use group-level values and prediction errors?

Line 380: report proportion missing trials instead of number?

Line 477: should link to table 3, not table 4?

Fig 4. Could you transform back the learning rate to a 0-1 range?

Line 570: were investigated, not where investigated

Reviewer #2 (Remarks to the Author):

The goal of the current study was to replicate previous effects (reported by Pessiglione et al.) of dopaminergic manipulations on reinforcement learning. While the previous dopaminergic effects on the learning rate could not be replicated, exploratory analyses suggest that increasing dopamine levels lowers the decision boundary in drift diffusion models. The manuscript is well written and addresses a timely topic that will be of interest to a broad readership working the neural basis of reward processing. However, there are also several limitations that hamper my enthusiasm.

First, the null effects of the pharmacological manipulation on the learning rates are interpreted as failed replication, but given the differences in task design (no loss condition) and administered doses compared with the original Pessiglione study one needs to be more cautious with interpreting these null results as failed replication. Second, previous studies already investigated the impact of dopamine on drift diffusion parameters, mainly reporting effects on the drift rate rather than the decision boundary (Beste et al., 2018; Westbrook et al., 2020; but also the authors' own work: Wagner et al., 2021, *Journal of Neuroscience*). This makes me wonder how paradigm-specific the current findings are and what they can tell us about the contribution of dopamine to decision making in general?

As the authors discuss themselves, the administered dose of haloperidol can have either presynaptic or postsynaptic effects in different individuals. It is thus possible that no mean effect of haloperidol on reward learning was observed because the presynaptic and postsynaptic effects cancelled each other out. The impact of a dose may depend on a participant's body weight or baseline dopamine levels (which are often estimated via performance in working memory tasks). Did the authors collect such measures? If yes, I ask them to add these variables as moderators of the individual drug effects to the statistical analyses. This would allow testing whether haloperidol affects learning rates depending on individual differences in body weight or working memory as proxy for baseline dopamine levels.

Participants performed the same learning task three times in separate experimental sessions. Did the authors test for learning/task repetition effects? Can they rule out that such task repetition effects contributed to the observed null effects? At least, the authors should control for the order of drug administration in their statistical analyses.

For the DDM, the authors report directional Bayes factors (BFs), and I am unsure how helpful these are for our interpreting the current findings. First, it seems more straightforward to compute the BFs in favor of the null over the alternative hypothesis, as the authors do for other analyses. Directional BFs, in contrast, indicate the strength of the evidence for an effect in the positive compared to the negative direction (rather than that the effect is zero), so it is not surprising that the reported directional BFs are relatively high, even for effects that are not “statistically significant” based on the 95% HDI. However, if it does not make sense to apply the established rules-of-thumb for BFs to the interpretation of directional BFs (Jarosz et al., 2014), I wonder whether they contribute at all to the interpretation of the current results.

Why was the starting point in the DDMs fixed to 0.5? Do models with such a fixed starting bias explain the data better than models with variable starting points?

Reviewer #3 (Remarks to the Author):

Summary:

This study investigated the role of dopamine in human reinforcement learning and decision making by examining the behavioral and neural effects of specific pharmacological modulations. The authors had two goals 1) Replicate Pessiglione 2006’s finding (albeit only using the “gain” condition) that L-DOPA (as opposed to Haloperidol) improved performance in a reinforcement learning task and 2) formalize L-DOPA’s role on decision making vs learning using the reinforcement learning drift diffusion model (RL-DDM). The main claims are that 1) L-DOPA did not improve learning, and there was no imaging evidence to support altered prediction errors but 2) L-DOPA and haloperidol both decreased the boundary separation, otherwise known as the decision threshold (formalized in the DDM) compared to placebo. The manuscript is well written and the authors’ analyses are thoughtful and appropriate; however the main conclusions require some more investigation.

Major issues:

- Despite no overall differences in accuracy, performance, or reaction time between drug conditions, the authors claim L-Dopa and haloperidol decrease the decision threshold. This result is potentially interesting and would in some sense imply improved performance if participants can maintain the same level of accuracy with lower threshold. However there are several concerns regarding this conclusion that would need to be further investigated before one can be confident in them.

- Changes in model parameters should always reflect some sort of change to behavior. Parameters might be more sensitive but one should always be able to see the features of the data that relate to the altered parameter, especially if the parameter is interpreted as a core result. While the authors conducted overall posterior predictive checks confirming the model accounted for the data, they did not confirm that the threshold changes were mediated by changes in behavior. This is important because DDM models can be mimicked with other variants of sequential sampling models in which other parameters might vary instead, so an interpretation in terms of threshold requires further validation.

- For example, a reduced threshold should lead to both speeded RTs - perhaps not at the median level but in the shape of the distribution - and to reduced accuracy in those cases where RTs are speeded. This can be examined via a quantile probability plot or speed accuracy tradeoff. The relation between threshold and median RT is not very convincing as it stands given that much of it is related to increases in RT for those subjects that have the smallest reductions in threshold...

- Moreover, the threshold reduction was only present in a subset of participants (Figure 5). The authors should test whether the participants who exhibit the strongest reductions in decision threshold (who are putatively sensitive to pharmacological manipulations) show greater changes in RT, accuracy, and imaging. The authors should also correlate the threshold reduction, not just to median

RT differences between sessions (Figure 5), but to the more specific differences predicted by a lower decision threshold in the selected participants. For example, when controlling for drift rate, a lower threshold should produce more errors and faster RTs in the slowest RT quantiles (distributions should be less skewed): were there differences between drug sessions when only looking at the slowest RT quantiles, and were these observed to a greater extent in those subjects who showed larger reductions in threshold? Alternatively, the magnitude of threshold reduction could relate to reduced slope of the learning curves. It would be helpful to do these analyses and also show posterior predictive checks for those subjects that have higher and lower thresholds, to see whether the model predicts these differences.

- The authors should further explore explanatory factors as to why some participants but not others exhibit a threshold decrease (as well as median RT differences between drug and placebo sessions). For instance, previous accounts have reported participant body weight predicting efficacy of fixed L-DOPA doses (see Rigoli et al., 2016; Figure 3 for an example of this relationship). Are there other between-subject factors that could account for these individual differences? Was it the same participants who were sensitive to both L-DOPA and haloperidol? Is there any other physiological evidence of drug status having an effect besides the reduction in decision threshold?

Given the main interpretation about drugs affecting choice parameters, it would be worthwhile to further examine whether drug status affects brain response (using the imaging data) at the time of choice. Although the authors explore chosen value activity, it would be sensible to also evaluate whether the striatal (or other regions) response varies as a function of the max (or average) Q value of the choice set (as predicted by Q learning algorithms) at choice onset. Does this relationship between Q value and neural response vary as a function of drug status, particularly in the subjects that are putatively sensitive to the boundary decrease effect? Similarly, it would be worthwhile to explore whether the brain response to the quantities that they do explore (chosen value and RPE) are altered to different degrees depending on the change in model parameter. See Schonberg 2010 for an example of this effect.

Minor

-L592: "might"

-L668: "pathways"

-Figure 6 should be reworked. The authors should clarify the overall group differences (or lack thereof) in imaging prediction errors statistics.

Response Letter for Chakroun et al.: Dopamine reduces decision thresholds in human reinforcement learning

Reviewer #1 (Remarks to the Author):

Reviewer #1: *In the current manuscript, Chakroun et al. present a study investigating the effect of L-Dopa and Haloperidol on a reinforcement learning task on 31 human participants in a blinded within-participants placebo-controlled study. fMRI data was collected while participants performed the task. Behavioural data was analysed using the reinforcement learning drift diffusion model.*

The analysis of behavioural and BOLD data did not replicate effects of increased reward prediction errors from L-Dopa vs. Haloperidol (Pessiglione et al., 06). Instead, the authors found that administration of both L-Dopa and Haloperidol lead to a decrease in decision threshold, a parameter from the RLDDM measuring how decision makers balance the speed-accuracy tradeoff.

I found this to be a very interesting manuscript focusing on an important topic, in which the analysis was performed soundly and the behavioural modeling steps were well described. However, I do have some questions and comments. My points are mainly focused on the computational model, and less so on the neurobiology.

Replication

The authors spend some time describing the study as a replication of Pessiglione et al., 06. However, the lack of replication is mainly interpreted as resulting from a difference in the dosage prescribed and the task design. But if that's the case, then why do you find effects for the decision threshold? It also makes the reader very curious to see whether the effect would indeed replicate in a more similar replication.

Response: Thanks for these comments, this raises an interesting point regarding the lack of replication. In short, we believe that a major difference between the performance and fMRI effects (where the findings from Pessiglione et al. (Pessiglione et al., 2006) did not replicate) and the decision threshold findings (where we observed consistent effects across drugs and a number of different modeling schemes) is that only in the former case, the lack of loss condition and reduced no. of symbols may have played a role. We now make this point more explicit in the discussion, where we now write (p. 13):

Notably, for learning-related effects, previously reported drug effects did not replicate (see discussion above), whereas for action selection, consistent reductions in decision thresholds were observed. Learning-related effects may be more affected by the omission of the loss condition than action-selection effects.

Reviewer #1: *Aims of paper. Related to the previous point, it is unclear whether the main goal was to replicate, or to discover whether dopamine is associated with reductions in decision threshold. From a critical perspective it could look like the discovery of a reduced decision threshold was used to change the focus from a replication study to a study investigating the effect of DA on cognitive mechanisms underlying learning and decision making. I don't think either approach is wrong, but that the initial goal of the study should be more clear.*

Response: Thanks for the opportunity to clarify. The initial goal of the study was to replicate. However, based on the methodological developments in the context of reinforcement learning DDMs since the study was conducted, we leveraged these innovations in combination with the present data set to address the novel issue of dopaminergic modulation of decision thresholds. We now re-phrased the final paragraph of the introduction to make this more explicit (p. 5):

Although replication was the initial goal of this project, recent developments of combined reinforcement learning drift diffusion models (Fontanesi et al., 2019; Miletić et al., 2020; Pedersen et al., 2017; Shahar et al., 2019; Wagner et al., 2022) (RLDDMs) allowed us to leverage this data set to examine dopamine effects on action selection. Thus, we jointly tested DA accounts of learning (replication analysis) and action selection (de-novo analysis) within the same data set.

Reviewer #1: *Model. I thought the modeling steps and results were very well described. The following comments are just thoughts on additional approaches that could be interesting. As described in Berke 18, the role of DA is proposed to influence decision threshold. But specifically, they propose that the effect could be an urgency signal or a reduction in threshold during the decision (collapsing bound model). This hypothesis could be tested using a newly developed method to incorporate other sequential sampling models than the DDM. It would be interesting to see whether the effect of L-Dopa and Haloperidol are better described as an increased collapse in threshold compared to a general reduction. A tutorial on applying such a model for an RL-task is described here:*

https://hddm.readthedocs.io/en/latest/demo_HDDMnnRL.html

Response: Thanks for this interesting point. We agree that the role of dopamine is likely complex, and might involve a general reduction in threshold and/or an effect on the degree of threshold collapse. We therefore followed the Reviewer's suggestion, and re-fit the data using the HDDM toolbox (version 0.9.8). Specifically, we compared a model with stable bounds (HDDMRL function) to a model with linearly collapsing bounds (HDDMnnRL function, 'angle' model). However, note two caveats to the results. First, the RLDDM with linearly collapsing bounds is presently only implemented with a single learning rate. Therefore, both models were only fit with a single learning rate, despite the fact that our model comparison revealed a superior fit of a model with dual learning rates. Second, we had to manually adjust some of the implemented parameter bounds, which were too narrow, e.g. with respect to the drift rate modulation. Generally, the data were better accounted for by the 'angle' model (DIC = 1320.59) compared to the standard model with fixed boundaries (DIC = 1646.04).

Response Letter Figure 1. An RLDDM with constant bounds implemented in the HDDM toolbox reproduced the drug effects on thresholds (α) obtained via JAGS. A: Boundary separation parameter per condition. B: Difference distributions (Haldol – Placebo and L-dopa – Placebo). Solid (thin) horizontal lines denote 85% (95%) highest posterior density intervals.

Response Letter Figure 2. Results from an RLDDM with linearly collapsing bounds implemented in the HDDM toolbox. A: The overall decision threshold (α) was reduced under both L-dopa and Haloperidol. B: Difference distributions for α (Haldol – Placebo and L-dopa – Placebo). C: Boundary collapse angle θ per drug. D: Difference distributions for θ (Haldol – Placebo and L-dopa – Placebo). Solid (thin) horizontal lines denote 85% (95%) highest posterior density intervals.

First, as can be seen from Response Letter Figure 1A and B, the HDDM standard model with stable bounds reproduced the drug effect on decision thresholds previously obtained via JAGS. Second, a threshold reduction of similar magnitude was still present in the collapsing bounds model (Response Letter Figure 2A, B). In contrast, there was no evidence for a difference in the degree of threshold collapse between conditions (Response Letter Figure 2C, D). Rather, numerically, threshold collapse was higher under placebo. Together, these results suggest that in our data, the data were more in line with DA impacting the overall decision threshold, rather than the collapse rate. We now added these both Figures to the Supplement, and refer to these analyses in the section on computational modeling (p. 9) where we now write:

Based on the idea that basal ganglia output might convey an urgency signal (Berke, 2018), behavioral data were additionally fitted with an RLDDM with linearly collapsing bounds, as implemented in the HDDM toolbox (Wiecki et al., 2013). In this model, both drugs were again associated with reduced overall decision thresholds but did not reliably impact the degree of threshold collapse (Supplemental Figures S14 and S15).

Reviewer #1: *It seems to me that theories of the effect of DA on choice could also be interpreted as having an influence on the rate of evidence accumulation through boosting go vs. no-go actions weights (e.g., Collins & Frank, 14), instead of having an effect on decision threshold. Could you provide more justification as to why boosting action weights should influence decision threshold and not drift rate?*

Response: This is a good point, thanks for raising this issue. Collins & Frank refer to the decision threshold in their section on RT effects. Here, they suggest that the *Act* value, the

weighted contrast between Go and NoGo action weights, directly regulates the decision threshold in a diffusion model, and our reasoning was based on this approach as well as the theoretical account of Berke (2018). However, others have reported that DA might also affect the drift rate (see also (Beste et al., 2018) and (Westbrook et al., 2020), as well as the corresponding comment by Reviewer #2). However, these studies both used Methylphenidate, a different drug. One further complicating factor is that in the RLDDM, only net “value evidence” is accumulated, i.e. the drift rate per trial is a function of the weighted difference in Q-values between options. An effect of DA on option-specific drift rates would in our view require a different modeling approach that estimates separate drift rates per option, and not just a difference measure. However, an extensive discussion of these issues would in our view go beyond the scope of the paper. Nevertheless, in the limitations section, we now reference the relevant work regarding a potential role of DA in regulating (aspects of) the drift rate (p. 15), and also point out that further work is required to determine the degree to which these effects depend on the paradigm and the specifics of the pharmacological manipulation:

Finally, under some conditions, dopaminergic manipulations can also affect evidence accumulation (Beste et al., 2018; Wagner et al., 2020), such as the impact of benefits and costs on choice (Westbrook et al., 2020). Further work is required to determine the degree to which these effects depend on the task or the specifics of the pharmacological manipulation.

Reviewer #1: *I found the effect of Haloperidol on learning rate for negative feedback to be interesting, as it seems to relate to a decrease in the no-go pathway. Could the authors provide some reflections on this effect?*

Response: This is an interesting point. Note that in the final model, the reduction in the negative learning rate under Haloperidol was substantially more pronounced compared to the previous version of the paper. However, in contrast to the boundary separation effects, which were also observed in the modeling scheme with separate models per condition, the negative learning rate effect did not show up in the separate conditions models (Supplemental Figure S9). However, we also explored whether the difference distribution between L-dopa and Haloperidol effects on the negative learning rate (i.e. the drug effects shown in Figure 3e). This difference distribution showed a substantial overlap with 0, indicating that there was no substantial evidence for the negative learning rate effects being different between the two drugs. For reasons of brevity in light of an already much expanded manuscript and extensive Supplement, we only added a short paragraph to the discussion, where we now write (p. 15):

Haloperidol also reduced the negative learning rate, although this effect was only observed in the combined model. This might be linked to a decrease in the *no go* pathway, and resonates with some earlier findings (Frank & O’Reilly, 2006; Jocham et al., 2014). However, another study did not observe changes in learning rate under Haloperidol, but here only a single learning rate was estimated, and no RLDDM modeling scheme was applied (Wächtler et al., 2020).

Reviewer #1: *I would be interested to see whether you find any associations in the BOLD data related to the effect on decision threshold.*

Response: We ran exploratory correlation analyses on drug effects reported in Figure 6 and Supplemental Figure S16. No effects were observed. However, we have decided not to include these additional analyses in the paper, given that in the course of the revisions, the Supplement has already been extended to include sixteen Supplemental Figures. We decided against running

additional whole-brain correlation analyses with individual threshold effects as predictors due to concerns of false positives, given the small sample size for such between-subjects analyses.

Reviewer #1: Minor *Line 333: could be confusing for some to use model-based (i.e. as in model-based vs. model-free). Maybe instead use model-derived?*

Response: This has been corrected.

Reviewer #1: *Line 355: why did you use group-level values and prediction errors?*

Response: Thanks for raising this issue. We followed the original study (Pessiglione et al., 2006) who also used group-level estimates for the computation of parametric regressors (although, in that case, a fixed effects maximum likelihood approach was used, rather than a hierarchical Bayesian approach). We now clarified this on p. 23:

Following Pessiglione et al (2006), values and prediction errors were calculated using the condition-specific group-mean learning rates of the best-fitting model RLDDM2 (see Table 5). All parametric regressors were z-scored within subject prior to entering the first level model⁸².

Reviewer #1: *Line 380: report proportion missing trials instead of number?*

Response: Thanks for this suggestion. We reported the number of missing trials rather than the proportion because our design suffers from rather low trial numbers per condition to begin with. Therefore, reporting proportions would be somewhat misleading, as a low proportion does not imply that final trial numbers are sufficient for modeling and fMRI analyses.

Reviewer #1: *Line 477: should link to table 3, not table 4?*

Response: This has been corrected.

Reviewer #1: *Fig 4. Could you transform back the learning rate to a 0-1 range?*

Response: We deliberately decided to leave the learning rates in the original space, and not backtransform them. The reason is that distributions and condition effects are much easier to visually appreciate this way. For example, negative learning rates are very small when backtransformed to the 0-1 range, such that the resulting distributions can push at the boundary. This can lead to the erroneous visual impression that learning rates are effectively zero (which they are not, as per e.g. Figure 3e). We therefore believe it is more helpful to leave learning rates and condition effects on learning rates in the original space, as to avoid such misconceptions.

Reviewer #1: *Line 570: were investigated, not where investigated*

Response: Thanks, the typo has been corrected.

Reviewer #2 (Remarks to the Author):

Reviewer #2: *The goal of the current study was to replicate previous effects (reported by Pessiglione et al.) of dopaminergic manipulations on reinforcement learning. While the previous dopaminergic effects on the learning rate could not be replicated, exploratory analyses suggest that increasing dopamine levels lowers the decision boundary in drift diffusion models. The manuscript is well written and addresses a timely topic that will be of interest to a broad readership working the neural basis of reward processing. However, there are also several limitations that hamper my enthusiasm.*

First, the null effects of the pharmacological manipulation on the learning rates are interpreted as failed replication, but given the differences in task design (no loss condition) and administered doses compared with the original Pessiglione study one needs to be more cautious with interpreting these null results as failed replication.

Response: Thanks for raising this issue. We absolutely agree, and have emphasized these differences throughout the discussion. We now re-phrased the abstract, and instead of talking about a “failed replication”, we now specify:

Previously reported beneficial effects of L-dopa vs. Haloperidol on reinforcement learning from gains and altered neural prediction error signals were not observed, which may be partly due to differences experimental design and/or drug dosages.

Second, previous studies already investigated the impact of dopamine on drift diffusion parameters, mainly reporting effects on the drift rate rather than the decision boundary (Beste et al., 2018; Westbrook et al., 2020; but also the authors’ own work: Wagner et al., 2021, Journal of Neuroscience). This makes me wonder how paradigm-specific the current findings are and what they can tell us about the contribution of dopamine to decision making in general?

Response: Thanks for raising this issue, which resonates with a point made by Reviewer #1. The potential paradigm-specificity is indeed an important point. But two further complicating factors need to be mentioned as well. First, our previous study (Wagner et al., 2020) adopted an arguably less sensitive between-subjects design, whereas other studies (Beste et al., 2018; Westbrook et al., 2020) as well as the present study used within-subjects designs. Second, the studies not only used different tasks (as mentioned by this Reviewer) but also different drugs. Beste et al. used two different dosages of MPH vs. placebo (in two separate groups), whereas Westbrook et al. used Methylphenidate vs. Sulpiride vs. Placebo in a single group. However, while Beste et al. (2018) report that Methylphenidate did not impact decision thresholds in a perceptual decision-making task, Westbrook et al. (2020) did not report how drugs impacted decision thresholds. On the other hand, our own recent work on tyrosine supplementation (Mathar et al., 2022) revealed reduced decision thresholds in two very different tasks, temporal discounting and a two-step sequential RL task, for tyrosine vs. placebo administration. This suggests that under some conditions, threshold effects can occur in very different task. Nonetheless, these effects require independent replication. Although we believe that an extensive discussion of these points is beyond the scope of the paper, we have added a section to the limitations section (p. 15) referencing the prior work on evidence accumulation and dopamine mentioned by this Reviewer. We also make the point that determining the degree to which these effects depend on the paradigm and drug manipulation require further work:

Finally, under some conditions, dopaminergic manipulations can also affect evidence accumulation (Beste et al., 2018; Wagner et al., 2020), such as the impact of benefits and costs

on choice (Westbrook et al., 2020). Further work is required to determine the degree to which these effects depend on the task or the specifics of the pharmacological manipulation.

Reviewer #2: As the authors discuss themselves, the administered dose of haloperidol can have either presynaptic or postsynaptic effects in different individuals. It is thus possible that no mean effect of haloperidol on reward learning was observed because the presynaptic and postsynaptic effects cancelled each other out. The impact of a dose may depend on a participant's body weight or baseline dopamine levels (which are often estimated via performance in working memory tasks). Did the authors collect such measures? If yes, I ask them to add these variables as moderators of the individual drug effects to the statistical analyses. This would allow testing whether haloperidol affects learning rates depending on individual differences in body weight or working memory as proxy for baseline dopamine levels.

Response: Yes, both working memory capacity and body weight were assessed, but we had not analyzed these data in the context of the present task so far. As noted in the companion paper (Chakroun et al., 2020) we assessed Listening Span, Rotation Span and Operation Span. Measures of Digit Span and Block Span were also obtained, but due to missing data in a number of participants who misunderstood the task instructions, these were omitted from the analysis. As suggested by Reviewer #2, we now included these measures in our analyses. As in Chakroun et al. (2020), we first used principal components analysis to combine measures into a single working memory capacity (WMC) score. The first component across tasks accounted for 52% of variance (see Response Letter Figure 3a), and exhibited positive coefficients for all working memory measures (Response Letter Figure 3b). Participant's scores for this first component were therefore taken as an individual difference measure of WMC.

Response Letter Figure 3. Principal component analysis results across all $n=31$ participants for three working memory measures: Listening Span (Lspan), Operation Span (OSpan) and Rotation Span (RSpan). a: Cumulative variance explained across the three principal components. The first component accounted for around 52% of variance. b: Component coefficients. The first principal component had positive loadings for all three working memory measures.

We felt reluctant to selectively add covariates *post hoc* for specific parameters that showed drug-effects in earlier analyses, such as decision threshold. At the same time, the Reviewer raises an important point, and these covariates might clearly play a role. We therefore set up a full model including all covariates for drug effects, separately for each parameter. Note that we included both linear and quadratic effects of WMC, given the extensive literature on non-linear dopaminergic drug effects (Cools & D'Esposito, 2011). In the full model, $\theta_{d,j}$ corresponds to the value of model parameter θ in subject j under drug d :

$$\theta_{d,j} = \theta_{Pl,j} + S_{\theta,d,j} + \beta_{WMC_{\theta,d}} WMC_j + \beta_{WMC^2_{\theta,d}} WMC_j^2 + \beta_{weight_{\theta,d}} weight_j$$

Here, $\theta_{Pl,j}$ corresponds to parameter θ in subject j in the placebo condition, and $S_{\theta,d,j}$ is the overall effect of drug d on θ in subject j . The additional terms correspond to the effects of the covariates:

$\beta_{WMC_{\theta,d}}$: linear effect of WMC on the effect of drug d on parameter θ

$\beta_{WMC^2_{\theta,d}}$: quadratic effect of WMC on the effect of drug d on parameter θ

$\beta_{weight_{\theta,d}}$: linear effect of weight on the effect of drug d on parameter θ

Despite the considerable added complexity, this model exhibited good convergence. Results are shown in Response Letter Figure 4 (linear effect of WMC), Response Letter Figure 5 (quadratic effect of WMC) and Response Letter Figure 6 (linear effect of body weight):

Response Letter Figure 4. Results for linear effects of working memory capacity (WMC) from the full model including covariates. Top row: L-dopa effects, Bottom row: Haloperidol effects.

Response Letter Figure 5. Results for quadratic effects of working memory capacity (WMC²) from the full model including covariates. Top row: L-dopa effects, Bottom row: Haloperidol effects.

Response Letter Figure 6. Results for linear effects of body weight from the full model including covariates. Top row: L-dopa effects, Bottom row: Haloperidol effects.

Although this analysis revealed some potentially suggestive effects (e.g. the effect of WMC on learning rate increases following Haloperidol tended to be positive, see Figure 4d, e). However, none of the effects were robustly different from zero. Since these effects might nonetheless be

of interest to some readers, we now included these results in the Supplemental Figures S11 – S13 and refer to these results on p. 9, where we now write:

As a control analysis, effects of working memory capacity (WMC, the first principal component of a principal component analysis across listening span, operation span and rotation span tasks, see Supplemental Figure S10) and body weight were included in the hierarchical model as modulators of the drug effects. None of the drug effects were reliably modulated by these covariates (see Supplemental Figures S11 – S13).

Reviewer #2: *Participants performed the same learning task three times in separate experimental sessions. Did the authors test for learning/task repetition effects? Can they rule out that such task repetition effects contributed to the observed null effects? At least, the authors should control for the order of drug administration in their statistical analyses.*

Response: Thanks for raising this issue. As noted in the paper, participants completed the three conditions in counterbalanced order, such that potential order effects were evenly distributed among the drug conditions. We did test for “meta-learning” across sessions, as suggested by this Reviewer. Performance as a function of session is depicted in Supplemental Figure S1, which shows that, if anything, a numerical decrease in total rewards obtained across sessions. The corresponding analysis is reported in the section “model-agnostic results”, and revealed moderate evidence ($BF_{01}=8.41$) against an effect of session. Also, note that we conducted a further control analysis that more directly corresponds to the between-subjects design employed by Pessiglione et al. (2006), where such meta-learning can be ruled out. Specifically, we compared overall rewards between L-dopa and Haloperidol, the primary behavioral outcome reported in Pessiglione et al. (2006), restricting the analysis to participants that received Haloperidol or L-dopa on the first session. We now emphasize in the respective section (“model-agnostic analyses”) that this shows that even when meta-learning can be completely ruled out, no overall drug effect on performance was observed:

This was also the case when restricting the analysis to only those subjects who received L-Dopa and Haloperidol on the first session ($t_{19}=-.943$, $p=.358$), which also shows that even in participants where meta-learning across sessions can be ruled out, no performance effect was observed.

Reviewer #2: *For the DDM, the authors report directional Bayes factors (BFs), and I am unsure how helpful these are for our interpreting the current findings. First, it seems more straightforward to compute the BFs in favor of the null over the alternative hypothesis, as the authors do for other analyses. Directional BFs, in contrast, indicate the strength of the evidence for an effect in the positive compared to the negative direction (rather than that the effect is zero), so it is not surprising that the reported directional BFs are relatively high, even for effects that are not “statistically significant” based on the 95% HDI. However, if it does not make sense to apply the established rules-of-thumb for BFs to the interpretation of directional BFs (Jarosz et al., 2014), I wonder whether they contribute at all to the interpretation of the current results.*

Response: Thanks for raising this issue. We absolutely agree that the type of comparison that is performed in these different analyses needs to be taken into account, i.e. whether a null model is compared to an alternative, or whether the evidence for positive vs. negative effects is compared.

We implemented the suggestion to compute Bayes Factors for the null vs. the alternative. In the case of nested models (such as here) this can be done via the Savage-Dickey (S-D) Density Ratio, where the ratio of the prior and posterior densities at zero is taken as a

Bayes Factor in favour of the alternative (i.e., in favour of the respective parameter value being different from zero). Although this ratio is straightforward to compute from the prior distributions and the posterior distributions obtained via MCMC, one caveat is that results can be highly sensitive to the choice of prior (Wagenmakers et al., 2010). The reason is that the S-D ratio is the ratio of two point densities – the prior density at a value of zero, and the posterior density at zero. But the prior density at zero decreases with increasing variance of the prior. This can lead to the counterintuitive result that the S-D method yields Bayes Factors in favour of a null model, despite the data being highly inconsistent with the null model. This conflict resonates with an ongoing discussion in Bayesian modeling: On the one hand, vague priors are often preferred, as they assign a similar likelihood to a large range of parameter space, and therefore ensure that the posterior is largely “unbiased” and dominated by the data. This is the reason why we used vague priors for drug effects throughout the combined models. At the same time, a consequence of using vague priors is that when they are combined with the S-D method to compute Bayes Factors, counterintuitive results can emerge (see Wagenmakers et al., 2010). For example, a parameter value of exactly zero can be more likely under the prior than the posterior, while at the same time, the posterior probability for a parameter value < 0 can exceed 97.5%. The latter was the case for for the decision threshold effects.

To illustrate this, and to quantify the dependency of drug effects on the prior, we systematically varied the precision of the prior. Drug effects were modeled using Cauchy priors with location parameter 0, and the scale parameters were systematically varied. Cauchy priors rather than Gaussian priors were now used to make these results more comparable to the Bayesian analyses conducted using JASP, which also uses Cauchy priors. Specifically, for drug effects on each parameter, the scale of the Cauchy prior was varied between .05 and 2.5 SD units, based on the estimated standard deviation of the respective parameter from RLDDM2 model fit to the placebo data. Results are shown in Response Letter Figure 7:

Response Letter Figure 7. Assessment of the impact of differences in prior specifications on drug-effects (orange: Haloperidol, red: L-dopa). Top row: directional Bayes Factors (dBF) quantifying the relative evidence for drug-induced increases vs. decreases for each parameter. Center row: Savage Dickey Density Ratio Bayes Factor in favour of a drug effect (BF10, dashed lines denote a BF10 of 3). Bottom row: posterior probability that the drug effect is < 0 , for each parameter (dashed lines denote a 95% probability that the effect is < 0).

As can be seen, there was in many cases a poor correspondence between the S-D Bayes Factor in favour of an effect (center row) and the posterior probability that an effect is < 0 computed based on the posterior distributions (bottom row). In contrast, the directional Bayes Factor

shows a much more consistent positive association with the probability that an effect is different from zero. To summarize, further modeling and simulation work is clearly required to arrive at better standards for these types of analyses.

For the purposes of the present report, we addressed this issue in the following way. We now first report the posterior probabilities for each drug effect being < 0 . Next, we report the 95% HDIs and their overlap with 0 and the ROPE. Finally, we also include the directional Bayes Factors, for completion, in case readers might find this informative.

Reviewer #2: *Why was the starting point in the DDMs fixed to 0.5? Do models with such a fixed starting bias explain the data better than models with variable starting points?*

Response: Thanks for raising this issue. Our approach of fixing the starting point at .5 was based on earlier modeling work using RLDDMs (Fontanesi et al., 2019; Pedersen et al., 2017; Shahar et al., 2019). Here, as in our work, boundaries were coded in terms of optimal vs. suboptimal choices (see Equation 4) and option allocation to the left vs. right side of the screen was randomized across trials. This implies that allowing a variable starting point in the model may be theoretically problematic, as it entails the assumption that participants have an *a priori* bias towards the “optimal” option, even before any learning has occurred. Note that this contrasts with DDM applications in the context of value-based decision-making tasks (Peters & D’Esposito, 2020), where an *a priori* preference for one option category (immediate or risky rewards, for example) is of course plausible.

However, it could also be argued that a bias towards optimal options might well emerge in later trials, following learning (Pedersen et al., 2017). We therefore implemented the model as suggested by this Reviewer, allowing the starting point to vary freely. Here, the starting point z was estimated in standard normal space (Uniform prior between -5 and 5 for the group mean), and back-transformed to the interval [0, 1] prior to calculation of the likelihood. To validate the model, we first focused on the placebo condition. However, in line with our considerations above, this model exhibited notoriously poor convergence for a number of individual-subject and group-level parameters (in particular group-level standard deviations, learning rates, and value coefficients of the drift rate). These convergence issues persisted when sampling was increased, first by a factor of 2, then 5 and then 10, i.e. up to 1.000.000 burn-in samples. Maximum \hat{R} values remained unacceptably high, i.e. > 7 . Note that this convergence issue already arose when restricting the analysis to the placebo condition data. Model convergence is typically an even greater issue when models including drug effects are estimated. For this reason, we refrained from further exploration of this model. These findings and considerations are now included in the methods section (p. 19) where we now write:

Following earlier work (Fontanesi et al., 2019; Mathar et al., 2022; Pedersen et al., 2017; Shahar et al., 2019; Wagner et al., 2022) and based on the assumption that participants do not have an *a priori* bias for optimal vs. suboptimal options prior to learning, z was fixed to .5. Because a bias might nonetheless develop over the course of learning (Pedersen et al., 2017), we also explored a model with variable starting point. However, this model failed to converge (maximum $\hat{R} > 7$), even when restricting the analysis on the data from the placebo condition, and when sampling was increased by a factor of ten. This model was therefore not explored further.

Reviewer #3 (Remarks to the Author):

Reviewer #3: *Summary: This study investigated the role of dopamine in human reinforcement learning and decision making by examining the behavioral and neural effects of specific pharmacological modulations. The authors had two goals 1) Replicate Pessiglione 2006's finding (albeit only using the "gain" condition) that L-DOPA (as opposed to Haloperidol) improved performance in a reinforcement learning task and 2) formalize L-DOPA's role on decision making vs learning using the reinforcement learning drift diffusion model (RL-DDM). The main claims are that 1) L-DOPA did not improve learning, and there was no imaging evidence to support altered prediction errors but 2) L-DOPA and haloperidol both decreased the boundary separation, otherwise known as the decision threshold (formalized in the DDM) compared to placebo. The manuscript is well written and the authors' analyses are thoughtful and appropriate; however the main conclusions require some more investigation.*

Major issues:

Despite no overall differences in accuracy, performance, or reaction time between drug conditions, the authors claim L-Dopa and haloperidol decrease the decision threshold. This result is potentially interesting and would in some sense imply improved performance if participants can maintain the same level of accuracy with lower threshold. However there are several concerns regarding this conclusion that would need to be further investigated before one can be confident in them.

Reviewer #3: – *Changes in model parameters should always reflect some sort of change to behavior. Parameters might be more sensitive but one should always be able to see the features of the data that relate to the altered parameter, especially if the parameter is interpreted as a core result. While the authors conducted overall posterior predictive checks confirming the model accounted for the data, they did not confirm that the threshold changes were mediated by changes in behavior. This is important because DDM models can be mimicked with other variants of sequential sampling models in which other parameters might vary instead, so an interpretation in terms of threshold requires further validation.*

Response: Thanks for raising this issue. We agree that the observed threshold effect might be reflected in other parameters when using different sequential sampling modeling frameworks. This also resonates with a point raised by Reviewer #1, who suggested that DA might also impact the threshold collapse over time in a collapsing bounds model (similar to an urgency signal), rather than the overall decision threshold (see Reviewer #1 section above). To address this point, we fitted additional models using the HDDM toolbox to test whether the drug effects were better accounted for by an overall reduction in thresholds, or by a change in threshold collapse. Two models were estimated using the HDDM toolbox, an RLDDM with a constant threshold (to reproduce our initial results using this toolbox) and a model with linearly decreasing thresholds over time ('angle' model). In the model with stable bounds (Response Letter Figure 1) and in the collapsing bounds model (Response Letter Figure 2), robust drug effects were restricted to a reduction in overall thresholds, whereas the rate of threshold collapse was unaffected. These analyses at least argue against the possibility that dopamine impacts threshold collapse, rather than overall boundary separation. The corresponding analyses are now included in the Supplement (Supplemental Figure S14 & S15). We refer to these analyses on p. 9, where we now write:

Based on the idea that basal ganglia output might convey an urgency signal (Berke, 2018), behavioral data were additionally fitted with an RLDDM with linearly collapsing bounds, as implemented in the HDDM toolbox (Wiecki et al., 2013). In this model, both drugs were again associated with reduced overall decision thresholds but did not reliably impact the degree of threshold collapse (Supplemental Figures S14 and S15).

Reviewer #3: – *For example, a reduced threshold should lead to both speeded RTs - perhaps not at the median level but in the shape of the distribution - and to reduced accuracy in those cases where RTs are speeded. This can be examined via a quantile probability plot or speed accuracy tradeoff. The relation between threshold and median RT is not very convincing as it stands given that much of it is related to increases in RT for those subjects that have the smallest reductions in threshold...*

– *Moreover, the threshold reduction was only present in a subset of participants (Figure 5). The authors should test whether the participants who exhibit the strongest reductions in decision threshold (who are putatively sensitive to pharmacological manipulations) show greater changes in RT, accuracy, and imaging.*

The authors should also correlate the threshold reduction, not just to median RT differences between sessions (Figure 5), but to the more specific differences predicted by a lower decision threshold in the selected participants. For example, when controlling for drift rate, a lower threshold should produce more errors and faster RTs in the slowest RT quantiles (distributions should be less skewed): were there differences between drug sessions when only looking at the slowest RT quantiles, and were these observed to a greater extent in those subjects who showed larger reductions in threshold? Alternatively, the magnitude of threshold reduction could relate to reduced slope of the learning curves. It would be helpful to do these analyses and also show posterior predictive checks for those subjects that have higher and lower thresholds, to see whether the model predicts these differences.

The authors should further explore explanatory factors as to why some participants but not others exhibit a threshold decrease (as well as median RT differences between drug and placebo sessions). For instance, previous accounts have reported participant body weight predicting efficacy of fixed L-DOPA doses (see Rigoli et al., 2016; Figure 3 for an example of this relationship). Are there other between-subject factors that could account for these individual differences? Was it the same participants who were sensitive to both L-DOPA and haloperidol? Is there any other physiological evidence of drug status having an effect besides the reduction in decision threshold?

Response: Thanks for these helpful suggestions to further explore behavioral correlates of the threshold effects. To address the points raised by the Reviewer, in a first step, we examined RT quantiles. Response Letter Figure 8 (below) shows group-level mean RTs for each drug condition and RT quantile (10th, 30th, 50th, 70th and 90th percentiles, see Lewandowsky & Farrell, 2018).

Response Letter Figure 8. Group-level mean RTs (solid lines, right y-axis) and accuracies (dashed lines, left y-axis) as a function of RT distribution quantiles.

As suspected by this Reviewer, RTs were numerically faster and accuracy was lower for the longer RTs (but also for the very first quantile). We next followed this up with the suggested analyses of individual differences. One challenge for these analyses is the fact that relatively few trials are available per subject and condition. To account for this, we split individual RT distributions into three quantiles (rather than five, as done for the group data), and computed the accuracy and mean RT for each quantile. We then repeated the individual difference analyses that was previously reported in the paper, this time focusing on the slowest third of trials, as suggested by Reviewer #3. In addition to examining associations between threshold-reductions and RT differences between conditions, we now also examined associations between individual differences in threshold reductions and individual differences in accuracy reductions (differences in arcsine sqrt transformed accuracy) between conditions for the slowest third of trials, as suggested by Reviewer #3. Results are shown in Response Letter Figure 9 and reported in the modified Figure 5 in the paper:

Response Letter Figure 9. Individual-difference analyses focusing on the slowest 1/3 of trials per participant and condition. a: Threshold reduction under L-dopa vs. RT difference between Placebo and

L-dopa. b: Threshold reduction under Haldol vs. RT difference between placebo and haldol. c: Threshold reduction under L-dopa vs. Accuracy difference between Placebo and L-dopa. d: Threshold reduction under Haldol vs. accuracy difference between Placebo and Haldol.

To account for the fact that we now have two predictors of interest (RT difference and accuracy difference) we replaced the correlation analysis included in the previous version of the paper with Bayesian linear regression analyses, predicting threshold reductions separately for L-dopa and haldol conditions. To account for the last point raised (potential effects of e.g. body weight on drug effects), these regression analyses now included covariates of body weight, as well as linear and quadratic effects of working memory capacity (WMC). WMC was quantified as the first principal component of a PCA across listening span, operation span and rotation span tasks (see response to Reviewer #2 above, Response Letter Figure 3). As suggested by visual inspection of Response Letter Figure 9, for L-dopa, threshold reductions were linked to both RT differences and accuracy differences (both $BF_{incl} > 100$), whereas, for haldol, threshold reductions were only linked to RT differences ($BF_{incl} > 10.000$) but not accuracy differences ($BF_{incl} < 1$). There was no evidence for an effect of any of the other covariates (linear and quadratic effects of WMC, linear effects of body weight). These results are now reported in the corresponding section of the results section (p. 9):

Finally, to link the modeling results back to individual differences in behavior, we examined the degree to which drug effects on boundary separation were associated with differences in RTs and accuracy between conditions, focusing on the slowest third of trials. Prediction of L-dopa effects on boundary separation using Bayesian linear regression (controlling for WMC, WMC^2 and body weight) revealed evidence for effects of RT differences (Figure 5a, $BF_{incl} = 290.191$) and accuracy differences (Figure 5c, $BF_{incl} = 107.038$). The corresponding analysis for haldol effects revealed only evidence for an effect of RT differences (Figure 5b, $BF_{incl} > 10.000$) but not accuracy differences (Figure 5d, $BF_{incl} < 1$).

Regarding the question of whether the same participants were sensitive to L-dopa and Haloperidol, this was not the case. The boundary separation effects of L-dopa and Haloperidol were not correlated ($r=.0083$). Likewise, the boundary separation effect of Haloperidol was not correlated with the effect of Haloperidol on negative learning rate ($r=.022$). However, one potential caveat to these analyses of individual differences in terms of parameters extracted from a hierarchical model is that the shrinkage imposed by the model might reduce potentially meaningful individual differences (Scheibehenne & Pachur, 2015).

Regarding additional physiological evidence for drug effects, please see our response to the next point, which revealed potentially interesting effects in an additional analysis of average value effects during the choice phase.

Regarding the question of additional explanatory factors that could account for the individual differences in drug effects, we also ran an additional hierarchical Bayesian model with parameter-specific covariates of WMC, WMC^2 and body weight (see response to Reviewer #2 above). Specifically, in this model, the effect of each drug on each parameter was additionally modulated by effects of WMC, WMC^2 and body weight. Posterior distributions of these covariate effects are shown in Response Letter Figure 4 (linear effect of WMC), Response Letter Figure 5 (quadratic effect of WMC) and Response Letter Figure 6 (linear effect of body weight), and results are now included in the Supplement (Supplemental Figure S11-S13). None of the drug effects of model parameter were robustly modulated by these covariates, and we now refer to these analyses on p. 9, where we now write:

As a control analysis, effects of working memory capacity (WMC, the first principal component of a principal component analysis across listening span, operation span and rotation span tasks, see Supplemental Figure S10) and body weight were included in the hierarchical model as modulators of the drug effects. None of the drug effects were reliably modulated by these covariates (see Supplemental Figures S11 – S13).

Reviewer #2 also requested posterior predictive checks for individual subjects as a function of decision threshold and/or decision threshold reductions under drug. Posterior predictive checks for individual subject data are already included in the Supplement, for example checks of the full RT distributions are provided in Supplemental Figures S3-S5. We now include a Table (see below) that provides an overview of which participants exhibited the minimum, median and maximum boundary separation parameters under placebo as well minimum, median and maximum drug effects on boundary separation:

Response Letter Table 1. Compilation of individual subjects showing minimum, median and maximum boundary separation parameters under placebo as well as minimum, median and maximum drug effects on boundary separation.

	Minimum	Median	Maximum
Boundary separation (placebo)	sub05	sub11	sub03
Boundary reduction (ldopa)	sub28	sub24	sub20
Boundary reduction (haldol)	sub05	sub12	sub10

Visual inspection of posterior predictive checks of the participants from Response Letter Table 1 suggests that RLDDM2 provided a good account of individual participant RT distributions, irrespective of the specific degree of boundary separation or the magnitude of drug effects on boundary separation.

Reviewer #3: *Given the main interpretation about drugs affecting choice parameters, it would be worthwhile to further examine whether drug status affects brain response (using the imaging data) at the time of choice. Although the authors explore chosen value activity, it would be sensible to also evaluate whether the striatal (or other regions) response varies as a function of the max (or average) Q value of the choice set (as predicted by Q learning algorithms) at choice onset. Does this relationship between Q value and neural response vary as a function of drug status, particularly in the subjects that are putatively sensitive to the boundary decrease effect? Similarly, it would be worthwhile to explore whether the brain response to the quantities that they do explore (chosen value and RPE) are altered to different degrees depending on the change in model parameter. See Schonberg 2010 for an example of this effect.*

Response: Thanks for raising this point. Based on this suggestion, we ran an additional first-level model, replacing the chosen-unchosen value parametric regressor for the choice phase with a parametric effect of mean Q-values across options. This turned out to be a very informative analysis. First, overall value effects in our a priori region of interest including vmPFC were much more pronounced for the average value regressor (see Response Letter Figure 5a below) compared to the original chosen vs. unchosen value regressor (Response Letter Figure 5b):

Response Letter Figure 5. Average Q-value effect (a) and chosen – unchosen value effect (b). Shown are main effects across drug conditions. Both effects survived correction for multiple comparisons across our a priori defined reward value region of interest.

We now include this analysis along with the original chosen – unchosen value analysis in the manuscript. However, as in the previous chosen – unchosen value analysis, effects of average value on BOLD responses in our a priori defined reward ROI did not vary as a function of drug.

Next, as suggested by Reviewer #3, we ran an exploratory whole-brain analysis testing whether drug status affected these average value effects. Given the similar effects of both drugs on decision thresholds, we ran T-tests (L-dopa & Haldol > Placebo and Placebo > L-dopa & Haldol). We are aware that such exploratory *post-hoc* analyses raise the danger for false positives. The threshold was thus adjusted to $p < .0001$ (*uncorrected*), for exploratory purposes, and we refrain from overinterpreting these results due to their exploratory nature (see below). Nonetheless, this analysis revealed two regions in which average value effects were elevated for drug vs. placebo, the left anterior insula (see Response Letter Figure 6a) and the ACC / pre SMA region (Response Letter Figure 6b):

Response Letter Figure 6. Exploratory whole-brain analysis results ($p < .0001$, uncorrected) testing for increased effects under L-dopa & Haloperidol vs. Placebo. a: Glass-brain results at $p < .0001$, uncorrected). This revealed effects in the left anterior insula (b) and the ACC / pre SMA region (c). Box plots depict single-subject parameter estimates extracted from group-level activation peaks.

Although these results are exploratory, they nonetheless resonate with a number of papers that have implicated the ACC / pre SMA region in decision threshold adjustments (Cavanagh et al., 2011; Domenech & Dreher, 2010; Rae et al., 2014). Due to the exploratory nature of this analysis, we only include it in the supplement. In the results section, we now write (p. 10):

Given that no drug effects were observed in our *a priori* defined region of interest, a final exploratory analysis was performed, testing for drug effects on average Q-value effects at an uncorrected threshold of $p < .0001$. This revealed higher effects under L-dopa and Haldol compared to placebo in left anterior insula and the dorsal anterior cingulate / pre SMA region (Supplemental Figure S9 and Supplemental Table S4).

We also now added a paragraph to the discussion, where we provide a very cautious interpretation of these results, given that they were obtained in an exploratory analysis. Here we write (p. 15):

Exploratory analyses implicated a circuit involving anterior insula and ACC / preSMA in the drug-induced decision threshold modulation (Supplemental Figure S9), such that average value effects in these regions were higher under l-dopa and haloperidol compared to placebo. Caution is warranted when interpreting these condition-dependent differences in parametric effects (Lebreton et al., 2019), in particular given their exploratory nature. Nonetheless, the dorsal anterior cingulate / pre-SMA region has previously been implicated in control adjustments (Shenhav et al., 2016), including decision threshold modulation (Cavanagh et al., 2011; Domenech & Dreher, 2010; Forstmann et al., 2008; Rae et al., 2014), and is densely interconnected with subcortical dopaminergic circuits (Haber & Knutson, 2010). A contribution of this circuit to the observed decision threshold adjustments under elevated dopamine would therefore be in line with these previous findings, but further confirmatory evidence is required.

Reviewer #3: *Minor -L592: “might”*

Response: This typo has been corrected.

Reviewer #3: *-L668: “pathways”*

Response: This typo has been corrected.

Reviewer #3: *-Figure 6 should be reworked. The authors should clarify the overall group differences (or lack thereof) in imaging prediction errors statistics.*

Response: Figure 6 was now split into two separate Figures, such that the prediction error effects are now reported in Figure 7. We also modified the description of the analysis on p. 10 to increase clarity. This section now reads (p. 10):

To reproduce the analysis from Pessiglione et al. (Pessiglione et al., 2006), positive and negative prediction error effects were extracted from bilateral ventral striatal regions that encoded model-derived prediction errors (see Figure 7a and Table 5) in GLM1. Using GLM2 that included separate predictors for positive and negative prediction errors, the corresponding parameter estimates were extracted (Figure 7b). While Pessiglione et al. (Pessiglione et al., 2006) reported a greater contrast between positive and negative striatal prediction errors under L-dopa compared to Haloperidol, this was not the case in our data. Bayesian ANOVAs with covariates of body weight, WMC and WMC² and the factors prediction error (positive vs. negative) and drug (Placebo vs. L-dopa vs. Haloperidol) only revealed evidence for a prediction error effect (left ventral striatum, $BF_{\text{incl}}=142.111$, right ventral striatum, $BF_{\text{incl}}=6.138$, all other $BF_{\text{incl}}<1.045$).

References

- Berke, J. D. (2018). What does dopamine mean? *Nature Neuroscience*, *21*(6), 787–793. <https://doi.org/10.1038/s41593-018-0152-y>
- Beste, C., Adelhöfer, N., Gohil, K., Passow, S., Roessner, V., & Li, S.-C. (2018). Dopamine Modulates the Efficiency of Sensory Evidence Accumulation During Perceptual Decision Making. *The International Journal of Neuropsychopharmacology*, *21*(7), 649–655. <https://doi.org/10.1093/ijnp/pyy019>
- Cavanagh, J. F., Wiecki, T. V., Cohen, M. X., Figueroa, C. M., Samanta, J., Sherman, S. J., & Frank, M. J. (2011). Subthalamic nucleus stimulation reverses mediofrontal influence over decision threshold. *Nat Neurosci*, *14*, 1462–1467.
- Chakroun, K., Mathar, D., Wiehler, A., Ganzer, F., & Peters, J. (2020). Dopaminergic modulation of the exploration/exploitation trade-off in human decision-making. *ELife*, *9*. <https://doi.org/10.7554/eLife.51260>
- Cools, R., & D’Esposito, M. (2011). Inverted-U-shaped dopamine actions on human working memory and cognitive control. *Biol Psychiatry*, *69*, e113-25.
- Domenech, P., & Dreher, J.-C. (2010). Decision threshold modulation in the human brain. *The Journal of Neuroscience: The Official Journal of the Society for Neuroscience*, *30*(43), 14305–14317. <https://doi.org/10.1523/JNEUROSCI.2371-10.2010>
- Fontanesi, L., Gluth, S., Spektor, M. S., & Rieskamp, J. (2019). A reinforcement learning diffusion decision model for value-based decisions. *Psychonomic Bulletin & Review*, *26*(4), 1099–1121. <https://doi.org/10.3758/s13423-018-1554-2>
- Forstmann, B. U., Dutilh, G., Brown, S., Neumann, J., von Cramon, D. Y., Ridderinkhof, K. R., & Wagenmakers, E.-J. (2008). Striatum and pre-SMA facilitate decision-making under time pressure. *Proceedings of the National Academy of Sciences of the United States of America*, *105*(45), 17538–17542. <https://doi.org/10.1073/pnas.0805903105>
- Frank, M. J., & O’Reilly, R. C. (2006). A mechanistic account of striatal dopamine function in human cognition: Psychopharmacological studies with cabergoline and haloperidol. *Behav Neurosci*, *120*, 497–517.
- Haber, S. N., & Knutson, B. (2010). The reward circuit: Linking primate anatomy and human imaging. *Neuropsychopharmacology*, *35*(1), 4–26.
- Jocham, G., Klein, T. A., & Ullsperger, M. (2014). Differential modulation of reinforcement learning by D2 dopamine and NMDA glutamate receptor antagonism. *The Journal of Neuroscience: The Official Journal of the Society for Neuroscience*, *34*(39), 13151–13162. <https://doi.org/10.1523/JNEUROSCI.0757-14.2014>
- Lebreton, M., Bavard, S., Daunizeau, J., & Palminteri, S. (2019). Assessing inter-individual differences with task-related functional neuroimaging. *Nature Human Behaviour*, *3*(9), 897–905. <https://doi.org/10.1038/s41562-019-0681-8>
- Mathar, D., Erfanian Abdoust, M., Marrenbach, T., Tuzsus, D., & Peters, J. (2022). The catecholamine precursor Tyrosine reduces autonomic arousal and decreases decision thresholds in reinforcement learning and temporal discounting. *PLoS Computational Biology*, *18*(12), e1010785. <https://doi.org/10.1371/journal.pcbi.1010785>
- Miletić, S., Boag, R. J., & Forstmann, B. U. (2020). Mutual benefits: Combining reinforcement learning with sequential sampling models. *Neuropsychologia*, *136*, 107261. <https://doi.org/10.1016/j.neuropsychologia.2019.107261>
- Pedersen, M. L., Frank, M. J., & Biele, G. (2017). The drift diffusion model as the choice rule in reinforcement learning. *Psychonomic Bulletin & Review*, *24*(4), 1234–1251. <https://doi.org/10.3758/s13423-016-1199-y>
- Pessiglione, M., Seymour, B., Flandin, G., Dolan, R. J., & Frith, C. D. (2006). Dopamine-dependent prediction errors underpin reward-seeking behaviour in humans. *Nature*, *442*(7106), 1042–1045. <https://doi.org/10.1038/nature05051>

- Peters, J., & D'Esposito, M. (2020). The drift diffusion model as the choice rule in intertemporal and risky choice: A case study in medial orbitofrontal cortex lesion patients and controls. *PLOS Computational Biology*, *16*(4), e1007615. <https://doi.org/10.1371/journal.pcbi.1007615>
- Rae, C. L., Hughes, L. E., Weaver, C., Anderson, M. C., & Rowe, J. B. (2014). Selection and stopping in voluntary action: A meta-analysis and combined fMRI study. *NeuroImage*, *86*, 381–391. <https://doi.org/10.1016/j.neuroimage.2013.10.012>
- Scheibehenne, B., & Pachur, T. (2015). Using Bayesian hierarchical parameter estimation to assess the generalizability of cognitive models of choice. *Psychonomic Bulletin & Review*, *22*(2), 391–407. <https://doi.org/10.3758/s13423-014-0684-4>
- Shahar, N., Hauser, T. U., Moutoussis, M., Moran, R., Keramati, M., NSPN consortium, & Dolan, R. J. (2019). Improving the reliability of model-based decision-making estimates in the two-stage decision task with reaction-times and drift-diffusion modeling. *PLoS Computational Biology*, *15*(2), e1006803. <https://doi.org/10.1371/journal.pcbi.1006803>
- Shenhav, A., Cohen, J. D., & Botvinick, M. M. (2016). Dorsal anterior cingulate cortex and the value of control. *Nature Neuroscience*, *19*(10), 1286–1291. <https://doi.org/10.1038/nn.4384>
- Wächtler, C. O., Chakroun, K., Clos, M., Bayer, J., Hennies, N., Beaulieu, J. M., & Sommer, T. (2020). Region-specific effects of acute haloperidol in the human midbrain, striatum and cortex. *European Neuropsychopharmacology: The Journal of the European College of Neuropsychopharmacology*. <https://doi.org/10.1016/j.euroneuro.2020.04.008>
- Wagenmakers, E.-J., Lodewyckx, T., Kuriyal, H., & Grasman, R. (2010). Bayesian hypothesis testing for psychologists: A tutorial on the Savage-Dickey method. *Cognitive Psychology*, *60*(3), 158–189. <https://doi.org/10.1016/j.cogpsych.2009.12.001>
- Wagner, B., Clos, M., Sommer, T., & Peters, J. (2020). Dopaminergic Modulation of Human Intertemporal Choice: A Diffusion Model Analysis Using the D2-Receptor Antagonist Haloperidol. *The Journal of Neuroscience: The Official Journal of the Society for Neuroscience*, *40*(41), 7936–7948. <https://doi.org/10.1523/JNEUROSCI.0592-20.2020>
- Wagner, B., Mathar, D., & Peters, J. (2022). Gambling Environment Exposure Increases Temporal Discounting but Improves Model-Based Control in Regular Slot-Machine Gamblers. *Computational Psychiatry*, *6*(1), Article 1. <https://doi.org/10.5334/cpsy.84>
- Westbrook, A., van den Bosch, R., Määttä, J. I., Hofmans, L., Papadopetraki, D., Cools, R., & Frank, M. J. (2020). Dopamine promotes cognitive effort by biasing the benefits versus costs of cognitive work. *Science (New York, N.Y.)*, *367*(6484), 1362–1366. <https://doi.org/10.1126/science.aaz5891>
- Wiecki, T. V., Sofer, I., & Frank, M. J. (2013). HDDM: Hierarchical Bayesian estimation of the Drift-Diffusion Model in Python. *Frontiers in Neuroinformatics*, *7*. <https://doi.org/10.3389/fninf.2013.00014>

REVIEWERS' COMMENTS

Reviewer #1 (Remarks to the Author):

I am very happy with the responses to my comments. Great work.
For the use of HDDMnnRL please cite (Pedersen & Frank, 21: <https://doi.org/10.1007/s42113-020-00084-w>) and (Fengler, Bera, Pedersen & Frank, 22: https://doi.org/10.1162/jocn_a_01902) in addition to Wiecki et al., 13

Reviewer #2 (Remarks to the Author):

The authors convincingly addressed all of my previous concerns on the manuscript.

Reviewer #3 (Remarks to the Author):

We thank the authors for thoroughly addressing most of our concerns. We think that the article is addressing an important and timely question about the role of dopamine in reinforcement learning and decision threshold modulation during action selection.

The manuscript is well written, the methods and analysis are thoughtful, and the main conclusions drawn are appropriate but should be carefully interpreted. We recommend this article for publication with the following minor revisions -

- The results section could incorporate results from the collapsing bounds model fit, especially in that it provided a better fit. Since there can be multiple accounts for reduced RTs or shorter RT tails, showing model fit with collapsing bound is important and would provide additional strength to the authors' argument about reduced decision thresholds as a result of pharmacological manipulations. The discussion section could have revisited this and briefly elaborated on this.
- On a related note, the more appropriate citation for the HDDMnnRL functionality within the HDDM toolbox would be Fengler et al. 2022 (J.CogNeuro), which is what enabled the fitting of collapsing bound and other models.
- For the sake of completeness, individual difference posteriors can be possibly incorporated into the Supplement.

Chakroun et al.: Dopamine reduces decision thresholds in human reinforcement learning

Response Letter 2nd Revision for

Reviewer #1 (Remarks to the Author): *I am very happy with the responses to my comments. Great work. For the use of HDDMnnRL please cite (Pedersen & Frank, 21: <https://doi.org/10.1007/s42113-020-00084-w>) and (Fengler, Bera, Pedersen & Frank, 22: https://doi.org/10.1162/jocn_a_01902) in addition to Wiecki et al., 13*

Response:

Thanks for this positive feedback. We appreciate the constructive comments. The suggested citations have been incorporated into the paper.

Reviewer #2 (Remarks to the Author): *The authors convincingly addressed all of my previous concerns on the manuscript.*

Response:

Thanks for this positive feedback. We appreciate the constructive comments.

Reviewer #3 (Remarks to the Author): *We thank the authors for thoroughly addressing most of our concerns. We think that the article is addressing an important and timely question about the role of dopamine in reinforcement learning and decision threshold modulation during action selection.*

The manuscript is well written, the methods and analysis are thoughtful, and the main conclusions drawn are appropriate but should be carefully interpreted.

Response: We thank the Reviewers for this positive assessment and the constructive feedback.

We recommend this article for publication with the following minor revisions –

- The results section could incorporate results from the collapsing bounds model fit, especially in that it provided a better fit. Since there can be multiple accounts for reduced RTs or shorter RT tails, showing model fit with collapsing bound is important and would provide additional strength to the authors' argument about reduced decision thresholds as a result of pharmacological manipulations. The discussion section could have revisited this and briefly elaborated on this.

Response: Thanks for this suggestion. Our re-analysis of the data with the collapsing bounds model that was suggested by Reviewer #1 mainly served the purpose of ruling out a potential alternative explanation of the effects - the pharmacological challenge might have affected the rate of threshold collapse, rather than (or in addition to) the overall threshold. The re-analysis with the collapsing bounds model argues against this account, as only the boundary separation offset, but not the degree of threshold collapse, were credibly affected by the drugs.

We agree that, generally, a more comprehensive analysis of the collapsing bounds model might provide additional insights. However, in the context of this particular data set, further comparisons are complicated by the fact that the collapsing bounds model (at least in the implementation that is currently publically available) only allows for the estimation of a single learning rate. In contrast, our model comparison revealed that dual learning rates substantially improve the degree to which RT changes over the course of learning can be recapitulated by the RLDDM (see in particular Figure 2, left panels). This complicates any comparison of the RLDDM with dual learning rates (used for all primary analyses in the paper) and the collapsing bounds model. Given the already extensive supplement, we believe that these

additional analyses, given that the conclusions that can be drawn from them are somewhat limited, are beyond the scope of the present paper.

Reviewer #3: - *On a related note, the more appropriate citation for the HDDMnnRL functionality within the HDDM toolbox would be Fengler et al. 2022 (J.CogNeuro), which is what enabled the fitting of collapsing bound and other models.*

Response: Thanks for this comment, the corresponding citation has been incorporated into the manuscript.

Reviewer #3: - *For the sake of completeness, individual difference posteriors can be possibly incorporated into the Supplement.*

Response: Thanks for this suggestion. We are reluctant to expand the supplement even further. But note that the full posterior distributions from all hierarchical models, including individual-participant posteriors and all corresponding summary statistics (means, STDs, CIs) are provided in the OSF repository (see <https://osf.io/8vzgh/>, all details are provided in the readme-PDF). In addition, individual differences for the core parameters of interest (drug effects on boundary separation parameters) are already included in the scatter plots of Figure 4 from the main text.